# Molecular Pathways Linking High-Fat Diet and PM_2.5_ Exposure to Metabolically Abnormal Obesity: A Systematic Review and Meta-Analysis

**DOI:** 10.3390/biom14121607

**Published:** 2024-12-16

**Authors:** Sagrario Lobato, Víctor Manuel Salomón-Soto, Claudia Magaly Espinosa-Méndez, María Nancy Herrera-Moreno, Beatriz García-Solano, Ernestina Pérez-González, Facundo Comba-Marcó-del-Pont, Mireya Montesano-Villamil, Marco Antonio Mora-Ramírez, Claudia Mancilla-Simbro, Ramiro Álvarez-Valenzuela

**Affiliations:** 1Departamento de Investigación en Salud, Servicios de Salud del Estado de Puebla, 603 North 6th Street, Centro Colony, Puebla 72000, Mexico; sagrariolobato@cejus.edu.mx; 2Clínica de Medicina Familiar con Especialidades y Quirófano ISSSTE, 27 North Street 603, Santa Maria la Rivera Colony, Puebla 72045, Mexico; 3Educación Superior, Centro de Estudios, “Justo Sierra”, Surutato, Badiraguato 80600, Mexico; vsalomon@cejus.edu.mx (V.M.S.-S.); nherrera@ipn.mx (M.N.H.-M.); claudia.mancilla@correo.buap.mx (C.M.-S.); ramal57@cejus.edu.mx (R.Á.-V.); 4Facultad de Cultura Física, Benemérita Universidad Autónoma de Puebla, San Claudio Avenue and 22nd South Boulevard, Ciudad Universitaria Colony, Puebla 72560, Mexico; facundo.comba@correo.buap.mx; 5Departamento de Medio Ambiente, Centro Interdisciplinario de Investigación para el Desarrollo Integral Regional Unidad Sinaloa, Instituto Politécnico Nacional, Juan de Dios Bátiz Boulevard 250, San Joachin Colony, Guasave 81049, Mexico; 6Facultad de Enfermería, Benemérita Universidad Autónoma de Puebla, 25th Avenue West 1304, Los Volcanes Colony, Puebla 74167, Mexico; 7Subsecretaría de Servicios de Salud Zona B, Servicios de Salud del Estado de Puebla, 603 North 6th Street, Centro Colony, Puebla 72000, Mexico; mireya.montesano@puebla.gob.mx; 8Instituto de Ciencias, Benemérita Universidad Autónoma de Puebla, San Claudio Avenue 1814, Ciudad Universitaria Colony, Puebla 72560, Mexico; marco.morar@correo.buap.mx; 9HybridLab, Fisiología y Biología Molecular de Células Excitables, Instituto de Fisiología, Benemérita Universidad Autónoma de Puebla, Prolongation of 14th South Street 6301, Ciudad Universitaria Colony, Puebla 72560, Mexico

**Keywords:** airborne particulate matter, high-fat diet, obesity, gene–environment interaction, oxidative stress, signaling pathways, metabolic pathways

## Abstract

Obesity, influenced by environmental pollutants, can lead to complex metabolic disruptions. This systematic review and meta-analysis examined the molecular mechanisms underlying metabolically abnormal obesity caused by exposure to a high-fat diet (HFD) and fine particulate matter (PM_2.5_). Following the PRISMA guidelines, articles from 2019 to 2024 were gathered from Scopus, Web of Science, and PubMed, and a random-effects meta-analysis was performed, along with subgroup analyses and pathway enrichment analyses. This study was registered in the Open Science Framework. Thirty-three articles, mainly case–control studies and murine models, were reviewed, and they revealed that combined exposure to HFD and PM_2.5_ resulted in the greatest weight gain (82.835 g, *p* = 0.048), alongside increases in high-density lipoproteins, insulin, and the superoxide dismutase. HFD enriched pathways linked to adipocytokine signaling in brown adipose tissue, while PM_2.5_ impacted genes associated with fat formation. Both exposures downregulated protein metabolism pathways in white adipose tissue and activated stress-response pathways in cardiac tissue. Peroxisome proliferator-activated receptor and AMP-activated protein kinase signaling pathways in the liver were enriched, influencing non-alcoholic fatty liver disease. These findings highlight that combined exposure to HFD and PM_2.5_ amplifies body weight gain, oxidative stress, and metabolic dysfunction, suggesting a synergistic interaction with significant implications for metabolic health.

## 1. Introduction

Obesity is a chronic and multifactorial disease characterized by a persistent imbalance in energy homeostasis [1]. It has become a significant challenge for public health, tripling its prevalence worldwide over the last five decades [2,3]. Obesity is classified based on the body mass index (BMI) and metabolic status [4], with obesity defined as BMI ≥ 30 kg/m^2^ and overweight as BMI of 25–29.9 kg/m^2^ [5,6]. Additionally, based on metabolic status, it is categorized into metabolically healthy obesity, metabolically abnormal obesity, or unhealthy obesity [7,8]. In addition to being a disease itself, obesity is a risk factor for all non-communicable chronic diseases and exacerbates some communicable diseases [9,10].

Environmental pollution, including air pollution, has been identified as a factor in the multifactorial causality of obesity due to the correlation observed between this condition and increased exposure to environmental pollutants [11,12]. In recent years, air pollution has been particularly concerning; it includes a complex mixture of particles and gasses with diverse chemical and physical compositions, originating from various sources and exhibiting spatial and temporal variability in toxicity [13,14]. Airborne particulate matter, especially fine particles ≤ 2.5 microns in diameter (PM_2.5_), is generated by anthropogenic activities and natural sources. It adversely affects human health, ecosystems, and visibility and contributes to climate change [15,16].

The prevalence of obesity and air pollution from PM_2.5_ have reached unprecedented levels [17,18]. In 2021, the World Health Organization (WHO) declared that air pollution is the leading global environmental health issue, prompting updated air quality guidelines recommending stricter limits on daily and annual exposure to PM_2.5_ [19]. Faced with this complex landscape, a study was conducted in a bioinformatics context using the Rothman causal model to analyze the impact of chronic PM_2.5_ exposure on the etiology of metabolically abnormal obesity [20]. Three new transcriptional signatures were reported: FAT-PM_2.5_-CEJUS, FAT-PM_2.5_-UP, and FAT-PM_2.5_-DN. FAT refers to the Spanish acronym for transcriptional adipogenic signature, CEJUS refers to the Spanish acronym for “Justo Sierra” Study Center, UP refers to up-regulated, and DN refers to down-regulated. These signatures exhibited a transcriptional regulation profile in adipocytes that was statistically similar under high-fat diet (HFD) intake and chronic PM_2.5_ exposure, affecting the peroxisome proliferator-activated receptor (PPAR) signaling pathway, small-molecule transport, adipogenesis gene pathway, cytokine–cytokine receptor interaction, and hypoxia-inducible factor 1 (HIF-1) signaling pathway [20].

The PPAR signaling pathway plays a crucial role in regulating lipid metabolism and adipogenesis through its three subtypes: PPARα (PPAR alpha), PPAR-β/δ (PPAR beta/delta), and PPARγ (PPAR gamma) [21,22]. PPARα regulates the clearance of circulating lipids and gene expression related to lipid metabolism in the liver and skeletal muscle [23,24]. At the same time, PPAR-β/δ is involved in lipid oxidation and cell proliferation [25,26], and PPARγ promotes adipocyte differentiation and glucose uptake [27,28].

These receptors are activated upon binding specific lipid ligands, forming heterodimers with the retinoid X receptor (RXR) and binding to retinoic acid response elements (AREs) in target gene promoters, thereby modulating gene transcription [29,30]. Exposure to PM_2.5_ significantly impacts this signaling pathway. In experimental mouse models, it has been shown that these particles inhibit the expression of PPARα and PPARγ in the liver. This inhibition is associated with negative regulation of these receptors and increased hepatic lipotoxicity [31]. Additionally, functional studies have demonstrated that PM_2.5_ induces adipogenesis by activating PPARγ in a 3T3-L1 preadipocyte differentiation model [32].

The relationship between PPARs and obesity is reflected in their ability to regulate critical metabolic processes. In the context of obesity, PPARγ facilitates adipose-tissue formation, while PPARα and PPAR-β/δ influence lipid metabolism and metabolic health. The disruption of these pathways by PM_2.5_ contributes to metabolic dysfunctions associated with obesity, as shown in recent studies [27,32].

The transport of small molecules is a biological pathway that includes various mechanisms. Among these are protein-mediated transport by the ATP-binding cassette (ABC) family and the assembly, remodeling, and clearance of plasma lipoproteins. The ABC transporter superfamily includes transmembrane proteins with diverse functions [33,34]. These proteins transport amino acids, lipids, inorganic ions, peptides, saccharides, metals, drugs, and proteins across cell membranes against concentration gradients, utilizing energy from ATP hydrolysis [35,36,37]. Additionally, ABC transporters are involved in intracellular compartmental transport [36,38]. Protein-mediated transport by the ABC family through the ABCA7-1 complex is essential for moving phospholipids and cholesterol out of cells [39,40]. This complex forms on the cell surface, where apolipoprotein A-I (ApoA1) is an acceptor of phospholipids and cholesterol, while ABCA7-1 facilitates their export to the plasma membrane [41,42]. The formation of the ABCA7-1 complex is crucial for remodeling plasma lipoproteins and maintaining lipid homeostasis [43,44].

In the context of the assembly, remodeling, and clearance of plasma lipoproteins, three essential functions are recognized: lipid transport mediated by chylomicrons, the endocytosis and degradation of low-density lipoproteins (LDL), and lipid transport by high-density lipoproteins (HDL) [45,46]. These functions are divided into assembly, remodeling, and clearance processes. In chylomicron-mediated lipid transport, a reaction occurs where the chylomicron, with an outer coat of apolipoproteins A (ApoA) and C (ApoC), is converted into a chylomicron remnant [47,48]. This process is facilitated by the enzyme lipoprotein lipase (LPL), coactivated by apolipoproteins ApoA5 and ApoC [49,50]. LPL stimulates the hydrolysis of triglycerides (TG) present in the chylomicron, releasing long-chain fatty acids and diacylglycerols [47,51].

The adipogenesis gene pathway is essential for the differentiation process of preadipocytes into mature adipocytes and comprises various regulatory elements [52,53]. This pathway includes eight main categories: inhibitors of the transition to adipocytes, transcription factors (TF) and modulators, growth factors and hormones, markers of differentiated adipocytes, miscellaneous elements, insulin-action genes, potential lipodystrophy-associated genes, and adipocyte-secreted products [54]. Among these elements, Pparα and CCAAT/enhancer-binding protein alpha (Cebpα) are key TFs in adipogenesis. Pparα (PPARα) and Cebpα (CEBPα) are essential for regulating genes that promote the differentiation of preadipocytes into adipocytes, playing crucial roles in lipid-metabolism regulation and adipose-tissue formation [52].

Fibroblast growth factor 21 (FGF21) is a regulator of adipogenesis. In models of obesity and type 2 diabetes, FGF21 is an antidiabetic and lipid-lowering agent. The Forkhead box protein O1 (FoxO1) negatively regulates the expression of this gene through *PPARα* in the liver [55]. Additionally, the tet methylcytosine dioxygenase 2 (TET2), a protein that converts 5-methylcytosine (5mC) into 5-hydroxymethylcytosine (5hmC), participates in the epigenetic regulation of adipogenesis. The overexpression of *TET2* in adipocytes influences the regulation of genes such as *Cebpb*, *Cebpa*, and *Pparg*, and its depletion in murine models inhibits adipocyte hypertrophy and protects against HFD-induced obesity [56].

Exposure to PM_2.5_ significantly alters the adipogenesis pathway. Studies with Nrf2^−/−^ mice showed that exposure to these particles decreases *PPARα* and increases *PPARγ*, suggesting a negative impact of PM_2.5_ on the balance of these critical receptors for adipogenesis and lipid metabolism [57]. Additionally, exposure of microglial cells to PM_2.5_ revealed adverse effects, such as reduced cell viability and structural damage, with *Cebpa* regulation mediated by differentially expressed long non-coding RNAs (lncRNAs) [58].

The cytokine–cytokine receptor pathway is organized into eight main groups that regulate inflammation and metabolism: chemokines, class I helical cytokines, tumor necrosis factor (TNF) family, transforming growth factor-beta (TGF-β) family, class II helical cytokines, IL-1-like cytokines, IL-17-like cytokines, and other unclassified cytokines [59]. These cytokines are soluble proteins or glycoproteins that act as essential regulators in immune and inflammatory processes, host adaptation, cell growth and differentiation, cell death, angiogenesis, and tissue repair to maintain homeostasis [60,61]. Cytokines exert their biological effects by binding to specific receptors on the surface of target cells. This binding activates intracellular signaling cascades that modulate gene expression and cellular responses [62,63]. The cytokine groups are subdivided into subgroups based on their structure and specific functions [64,65].

The role of heterogeneous nuclear ribonucleoprotein A1 (HNRNPA1) in lipid and glucose metabolism has been investigated. Obesity-related studies in mice have observed downregulation of the *Hnrnpa1* gene in white adipose tissue (WAT). This decrease in *Hnrnpa1* promotes greater macrophage infiltration and an increase in the expression of proinflammatory and fibrosis genes, exacerbating insulin sensitivity, glucose intolerance, and hepatic steatosis. Additionally, *Hnrnpa1* has been found to regulate the stability of *Ccl2* mRNA, and its inhibition improves inflammation in WAT and glucose homeostasis [66]. Other studies have demonstrated that PM_2.5_ directly influences inflammation in visceral adipose tissue, alters fat metabolism in the liver, and affects glucose metabolism in skeletal muscle through both CCR2-dependent and independent mechanisms [67,68].

Research conducted in humans explored various approaches to address metabolic health and obesity, revealing significant findings about the cytokine–cytokine receptor pathway. A randomized controlled study with 55 students examined the impact of moderate jogging on PM_2.5_-induced high blood pressure. The results showed that this physical activity increased the levels of IL-6 and the myokine clustering and significantly reduced systolic blood pressure and inflammatory markers compared to the control group [69]. Another study investigated the effect of calorie restriction on walking speed, considering BMI and plasma IL-6 levels. This study found that while calorie restriction improved walking speed, especially in individuals with obesity and high IL-6 levels, the reduction in BMI was the main factor for this improvement, and changes in IL-6 levels did not significantly impact walking speed [70].

A study in Romania identified genetic loci associated with obesity. It found that 34.6% of adults were overweight and 31.4% were obese, with a high prevalence of cardiometabolic complications. This study revealed that the AG genotypes of the leptin (LEP) A-2548G polymorphism and the AA genotypes of the fat-mass and obesity-associated protein (*FTO*) rs9939609 polymorphism were associated with a higher risk of obesity. Additionally, the GRGMLA haplotype might be a susceptibility factor for obesity, with significant associations between *LEP* and leptin receptor (*LEP-R*), *LEP* and ghrelin (*GHRL*), and *GHRL* and *FTO* [71]. Another study examined DNA methylation levels in the nuclear respiratory factor 1 (*NRF1*), *FTO*, and *LEPR* genes in the saliva of children. The study revealed that overweight or obese Euro-American children had higher methylation in *NRF1* and *FTO*. In contrast, in African-American children, higher methylation of the *LEPR* gene was associated with average weight and showed a negative relationship with obesity measures [72].

The HIF-1 signaling pathway is a cellular mechanism activated in response to hypoxia. HIF-1 is a TF that regulates the expression of various genes involved in cellular adaptation to hypoxia [73,74]. This signaling begins with stabilizing and activating the HIF-1 complex, composed of the HIF-1α and HIF-1β subunits. The former is inducible, while the latter is constitutively expressed [73,75]. In normoxia, HIF-1α undergoes hydroxylation at specific proline residues, leading to its ubiquitination and subsequent degradation by the enzyme prolyl hydroxylase [73,76].

In the presence of hypoxia or other stimuli, such as nitric oxide or various growth factors, IL-6 binds to the IL-6R receptor on the cell surface, forming an active signaling complex [77,78]. This complex triggers an intracellular signaling cascade that activates STAT proteins, which induce the expression of the *Hif1a* gene, encoding the HIF-1α subunit [79,80]. The activity of PHD is reduced, thereby stabilizing HIF-1α, which then interacts with coactivators such as p300/CBP to modulate its transcriptional activity [81,82].

Under hypoxic conditions, the HIF-1α subunit translocates to the nucleus, where it binds to HIF-1β to form the active HIF-1 complex [75,83]. This complex acts as a transcription factor that binds to specific DNA sequences known as hypoxia-response elements (HREs), located in the promoter regions of genes encoding proteins involved in the hypoxic response [84,85]. These genes include those involved in angiogenesis, glycolysis, cell survival, and apoptosis [86,87].

The effect of HIF-2α on insulin secretion in β-cells of mice has been investigated, revealing that metabolic stress induces the activation of HIF-2α, which protects against mitochondrial damage caused by reactive oxygen species (ROS) [88]. It has also been proposed that, in obesity, metabolic dysregulation activates HIF-1α in adipose tissue macrophages, promoting chronic inflammation and insulin resistance. In obese mice, an increase in HIF-1α, IL-1β, and glycolytic genes was observed. At the same time, the deletion of HIF-1α reduced macrophage accumulation and IL-1β production, highlighting its role in regulating metabolic stress and inflammation in obesity [89].

This review hypothesizes that chronic exposure to an HFD and PM_2.5_ promotes a chronically oxidative cellular environment. This condition dysregulates the transcription of genes related to small molecule transport, PPAR and HIF-1 signaling pathways, cytokine–cytokine receptor interactions, and adipogenesis pathways through oxidative modification of transcription factors and DNA methylation enzymes. This study aims to perform a systematic review and a meta-analysis to provide a comprehensive overview of the molecular mechanisms involved in metabolically abnormal obesity induced by HFD and PM_2.5_ exposure. This study establishes a solid foundation for future research and therapeutic strategies in this field.

## 2. Materials and Methods

The study was carried out according to the preferred reporting items for systematic reviews and meta-analyses (PRISMA) guidelines [90] and registered in Open Science Framework (OSF) under the doi: 10.17605/OSF.IO/F2VGJ (https://osf.io/f2vgj).

### 2.1. Search Strategy

An exhaustive search was conducted using Scopus, Web of Science, and PubMed databases. The words used were “PM_2.5_” and “High-fat diet” in the fields “Title”, “Abstract”, “Keywords”, and “Topic” (Appendix A). The search range was from the start day of each database until 14 July 2024. The references obtained were exported to the reference manager ENDNOTE, where duplicate references were eliminated through an automatized process followed by a manual revision. Then, the full text of selected articles was collected.

### 2.2. Selection Criteria

Articles published in the last five years (2019–2024) that reported molecular pathways in their results were included, excluding those assessed as low quality according to the criteria established in Section 2.4, Quality Assessment.

### 2.3. Data Extraction

Data were collected in Microsoft Excel 365^®^. The results presented include study design, population or biological model characteristics, sample size, tissue or cell line, statistically significant metabolic parameters, and molecular mechanisms. Six researchers extracted and compared the data to ensure accuracy.

### 2.4. Quality Assessment

The quality of the articles in the meta-analysis was evaluated using specific tools according to the research design. For cohort and case–control studies, the Newcastle–Ottawa scale (NOS) was applied, which evaluated the selection of groups, their comparability, and the determination of exposure or outcome [91,92,93]. Cross-sectional studies were assessed with the Appraisal Tool for Cross-Sectional Studies (AXIS tool) [94,95,96]. The scores were interpreted as follows: for the NOS scale, the studies were rated as high quality (≥7 points), moderate (3–6 points), and low quality (<3 points) [97,98], while the AXIS tool assessed studies as high quality (>80%), moderate (60–80%) and low (<60%) [99,100]. The evaluation of the quality of the articles was conducted by four researchers (S.L., V.M.S., R.A., and M.N.H.). Discrepancies in qualification were resolved through consensual discussions among evaluators.

### 2.5. Statistical Analysis

The statistical analysis was performed using the software OpenMeta [Analyst] for Windows 10 [101,102,103]. The mean difference was selected in a random-effects meta-analysis model [104] with a 95% confidence interval (CI) [102]. The index of each study was calculated, and the results were combined to obtain an overall effect size. Heterogeneity was estimated using the inconsistency index (*I*^2^), with values above 25%, 50%, and 75% considered low, moderate, and high heterogeneity, respectively [105]. Subgroup analyses were conducted to resolve heterogeneity [104].

### 2.6. Biological Pathway Enrichment Analysis

The fold changes (FC) values of the transcripts extracted from the articles were standardized. Those values transformed to logarithms were reversed using the exponential function and the natural logarithm in a spreadsheet of Microsoft Excel 365^®^ [106] to enable their biological pathway enrichment analysis on the WEB-based GEne SeT AnaLysis Toolkit platform. [107]. This procedure followed the method described in our previous study [20]. Statistical significance was assessed with *p* ≤ 0.05 and a false discovery rate (FDR) ≤ 0.05. The schematics of each metabolic pathway were obtained from biological repositories [108,109,110].

## 3. Results

### 3.1. Selected Studies

Figure 1 shows the PRISMA flow diagram of the strategy to identify and select articles. After removing duplicates, we identified 47 studies from scientific databases, of which 14 were excluded because they were published before 2019 (n = 11) or did not report biological pathways in the results (n = 3). The number of articles included in this review and meta-analysis was 33 [20,111,112,113,114,115,116,117,118,119,120,121,122,123,124,125,126,127,128,129,130,131,132,133,134,135,136,137,138,139,140,141,142].

The data from the selected articles are summarized in Table 1. Of the 33 included studies, 88% followed a case–control design, and 94% utilized murine models as the primary biological system. Human studies represented 6% of the total: one involved human cell lines with a case–control design, and the rest were cross-sectional studies conducted on human participants. The sample sizes varied widely, with a mean of 32 and a standard deviation of 23.7. Hepatic and hematological tissues were the most frequently analyzed, appearing in 21.2% of the studies.

### 3.2. Body Weight According to Exposure to HFD and PM_2.5_

Figure 2 illustrates the estimated average body-weight gain in grams (g) at a general level and specific results of the subgroups of mice exposed to HFD, PM_2.5,_ and their combination. On a global level, the estimated average body-weight gain was 76.270 g (CI 95%: 49.880 to 102.661 g, *p* < 0.001), with a considerably high heterogeneity (*I*^2^ = 100%). In the control group, the gained weight was 69.579 g (CI 95%: 23.792 to 115.365 g, *p* = 0.003). The results were not statistically significant in the murine group fed with HFD.

On the other hand, exposure to PM_2.5_ generated a gain of weight of 77.077 g (CI 95%: 15.884 to 138.270 g, *p* = 0.014). The combination of HFD and PM_2.5_ produced a greater gain of weight estimated at 82.835 g (CI 95%: 0.631 to 165.039 g, *p* = 0.048). It is essential to highlight that all subgroups presented high heterogeneity, with values of *I*^2^ higher than 99%, which indicates a remarkable variability in the results between the analyzed studies.

### 3.3. Metabolic Biomarkers Induced by HFD and PM_2.5_

Figure 3 presents changes in different metabolic biomarkers, including global and subgroups, after exposure to HFD. At the general level, it was observed an average increase of 9.23 in the biomarkers (CI 95%: 6.334 to 12.137, *p* < 0.001), with high heterogeneity (*I*^2^ = 100%). In the analysis of subgroups, the changes in adiposity, the glucose tolerance test (GTT), and the insulin resistance index (HOMA-IR) were not statistically significant. In contrast, increases in the levels of total cholesterol, HDL, LDL, and TG were observed, with estimations of 8.98 (CI 95%: 8.58 to 9.37, *p* < 0.001), 8.60 (CI 95%: 7.86 to 9.35, *p* < 0.001), 6.77 (CI 95%: 3.67 to 9.88, *p* < 0.001), and 8.08 (CI 95%: 3.03 to 13.13, *p* = 0.002), respectively. The heterogeneity observed was null for total cholesterol (*I*^2^ = 0%), moderate for HDL (*I*^2^ = 49.9%), and high for LDL (*I*^2^ = 86.6%) and TG (*I*^2^ = 93.6%), which suggests that exposition to HFD causes significant changes in the lipidic profile, with notable variability in the response of LDL and TG between the included studies.

Furthermore, increases in glucose levels, insulin, and superoxide dismutase (SOD) were estimated, with values of 7.82 mg/dL (CI 95%: 0.86 to 14.78, *p* = 0.028), 5.02 ng/dL (CI 95%: 0.17 to 9.88, *p* = 0.042), and 11.79 U/mg protein (CI 95%: 2.01 to 21.58, *p* = 0.018), respectively. Glucose and insulin did not show heterogeneity (*I*^2^ = 0%), while the SOD presented high heterogeneity (*I*^2^ = 99.96%), which suggests a uniform response in the biomarkers of metabolism carbohydrates, while SOD presented considered variability in its antioxidant response between the different studies.

Figure 4 presents the changes in different metabolic parameters after the exposition to PM_2.5_ globally and in subgroups. At a general level, an average increase in the biomarkers of 9.16 (CI 95%: 8.50 to 9.81, *p* < 0.001) was observed, with a very high heterogeneity (*I*^2^ = 99.5%). The malondialdehyde (MDA) changes in the subgroups, the GTT, and the HOMA-IR were not statistically significant. In contrast, the SOD levels showed increases of 11.75 (CI 95%: 2.89 to 20.61, *p* = 0.009) with high heterogeneity (*I*^2^ = 99.4%), which indicates a significant response in the antioxidant systems because of the exposition to fine particulate matter, accompanied by a considerable variability between studies.

Total cholesterol, HDL, LDL, and TG levels also increased after exposition to PM_2.5_. The estimations were 7.63 mg/dL (CI 95%: 4.92 to 10.34, *p* < 0.001) for total cholesterol, 8.90 mg/dL (CI 95%: 8.63 to 9.16, *p* < 0.001) for HDL, 6.65 mg/dL (CI 95%: 2.71 to 10.59, *p* < 0.001) for LDL, and 9.08 mg/dL (CI 95%: 5.26 to 12.91, *p* < 0.001) for TG. The heterogeneity was low for the total cholesterol (*I*^2^ = 14.7%) and HDL (*I*^2^ = 38.2%), in contrast with LDL (*I*^2^ = 93.2%) and TG (*I*^2^ = 98.4%), which showed high heterogeneity. This suggests a significant increase in the lipid profile after exposition to PM_2.5_, particularly in the levels of HDF and TG. Furthermore, the levels of glucose and insulin increased, with values of 7.70 (CI 95%: 0.88 to 14.52, *p* = 0.027) and 5.139 (CI 95%: 3.25 to 7.02, *p* < 0.001), respectively, without showing heterogeneity (*I*^2^ = 0%). These results reveal a consistent response in the increase of these biomarkers after exposition to PM_2.5_, suggesting a significant impact on glucose metabolism.

Figure 5 presents the changes in diverse metabolic biomarkers for combined exposition to HFD and PM_2.5_. On a general level, an average increase in the biomarkers of 3501.46 (CI 95%: 3347.52 to 3655.41, *p* < 0.001) is observed, accompanied by extremely high heterogeneity (*I*^2^ = 100%). Although some subgroups showed significant increases, others, such as the MDA, the GTT, HOMA-IR, glucose, and total cholesterol, did not reach statistical significance. The concentrations of HDL, LDL, and TG showed increases, with estimations of 8.94 (CI 95%: 8.75 to 9.13, *p* < 0.001), 6.87 (CI 95%: 3.36 to 10.37, *p* < 0.001), and 8023.334 (CI 95%: 1702.11 to 14,344.55, *p* < 0.001), respectively. A high heterogeneity for total cholesterol and TG (*I*^2^ = 100%) was observed, as for LDL (*I*^2^ = 86.7%), but low heterogeneity was detected for HDL (*I*^2^ = 21.3%), which indicates an increase in the lipidic profile, with notable variability in the studies except for HDL.

In addition, the levels of insulin and SOD showed increases, with values of 6.293 (CI 95%: 3.153 to 9.432, *p* < 0.001) and 12.68 (CI 95%: 3.70 to 21.65, *p* = 0.006), respectively. The heterogeneity was null for insulin (*I*^2^ = 0%) but high for SOD, with an *I*^2^ of 99.9%. These results reflect a significant alteration in the metabolism of the insulin and the antioxidant system for the combined exposition to both factors, with a consistent response between the studies in the insulin.

### 3.4. Biological Pathway Induced by HFD and PM_2.5_

Figure 6 shows the enrichment analysis results of gene groups (GSEA) and the over-presentation (ORA) of obtained transcripts from the articles, classified by tissues according to the exposition to HFD, PM_2.5_, and the combination of both factors. The details about biological repertories, transcripts, and evaluated tissues are available in Appendix A. In brown adipose tissue (BAT), the transcriptional pathways with statistical significance (*p* < 0.05 and FDR < 0.05) included the adipocytokine signaling pathway, with an enrichment ratio of 37.25, composed of three transcripts differentially expressed after exposure to HFD, and the adipogenesis genes, with an enrichment ratio of 12.88, integrated by four transcripts regulated in response to PM_2.5_ exposure (Figure 6A).

In WAT, both exposure to HFD and PM_2.5_ affected the same pathway, the metabolism of proteins, with a normalized enrichment score of −1.98 for HFD and −1.93 for PM_2.5_ (*p* < 0.05 and FDR < 0.05), each composed of three downregulated transcripts (Figure 6B). In cardiac tissue, exposure to HFD and the combination of HFD and PM_2.5_ activated common pathways: cellular responses to stress, cellular responses to stimuli, and cellular response to chemical stress, with a normalized enrichment score of −1.82 for HFD and −1.92 for the combined exposure (*p* < 0.05 and FDR < 0.05), each composed of three downregulated transcripts. However, in this tissue, exposure to PM_2.5_ affected the burn-wound healing pathway, with a normalized enrichment score of 1.71 (*p* < 0.05 and FDR < 0.05), composed of three upregulated transcripts (Figure 6C).

The significant transcriptional pathways (*p* < 0.05 and FDR < 0.05) in liver tissue include the PPAR signaling pathway, with an enrichment ratio of 36.03 and three transcripts that presented altered expression levels after exposure to HFD. Furthermore, an enrichment ratio of 42.7 in the same pathway was observed with four transcripts affected by exposure to PM_2.5_ and combining both factors. The AMP-activated protein kinase (AMPK) signaling pathway presented an enrichment ratio of 25.87, with three transcripts regulated by HFD, and an enrichment ratio of 22.99, with three transcripts affected by PM_2.5_ and the combination of both factors. Finally, non-alcoholic fatty liver disease showed an enrichment ratio of 21.31, with three transcripts regulated by HFD, and 18.94, with three transcripts affected by PM_2.5_ and the combination of both factors (Figure 6D).

#### 3.4.1. Adipocytokine Signaling Pathway in Brown Adipose Tissue

The adipocytokine signaling pathway (Figure 7), cataloged in KEGG with the code mmu04920, is metabolic and inflammatory. The increased volume and number of adipocytes are directly associated with increased leptin production and decreased adiponectin. Leptin, a crucial regulator of energy intake and metabolic rate, acts primarily in the hypothalamic nuclei. The anorexigenic effect of leptin occurs through the modulation of neuropeptides such as neuropeptide Y (NPY), agouti-related protein (AGRP), and alpha-MSH. This process is mediated by JAK kinase and STAT3 phosphorylation, resulting in nuclear transcriptional regulation.

On the other hand, adiponectin contributes to the reduction of plasma glucose levels and free fatty acids (FFA). This effect is due to the adiponectin-mediated activation of AMPK, which stimulates fatty acid oxidation in skeletal muscle and improves glucose uptake. Furthermore, activation of AMPK by adiponectin also suppresses endogenous glucose production by inhibiting the expression of PEPCK and G6Pasa.

TNF-alpha (TNFα) links obesity and insulin resistance by interfering with the early stages of insulin signaling. TNFα inhibits the phosphorylation in tyrosine of the IRS1 protein and promotes its phosphorylation in serine. Among the serine/threonine kinases activated by TNFα, JNK, mTOR, and IKK have been identified as key in this process, mediating the effects of TNFα on insulin signaling and metabolic regulation.

Based on the results, the transcripts analyzed (*Tnf*, *Lepr*, and *Pparα*) partially participate in the adipocytokine signaling pathway. These transcripts dysregulated TNFα, leptin receptors, and PPARα, altering insulin signaling, leptin function, and carnitine palmitoyltransferase 1 (CPT1) activity. This alteration restricts the transfer of long-chain fatty acids across the mitochondrial membrane, preventing their oxidation.

#### 3.4.2. Adipogenesis Genes in Brown Adipose Tissue

Figure 8 presents the pathway of genes involved in adipogenesis, according to the WikiPathways entry WP447. This pathway illustrates the elements involved in adipogenesis, the process by which preadipocytes differentiate into mature adipocytes. This pathway is composed of eight categories of elements: inhibitors of the transition of preadipocytes to adipocytes, TFs and modulators, growth factors and hormones, markers of fully differentiated adipocytes, miscellaneous elements, genes related to insulin action, possible genes associated with lipodystrophy, and products secreted by adipocytes. *Cebpa*, *Pparα*, *Serpine1*, and *Ucp1* are critical components of this pathway in the transcripts analyzed.

*Cebpα* and *Pparα* participate in the TF and modulators involved in the adipogenesis pathway. Both regulate the expression of genes essential for the differentiation of preadipocytes into mature adipocytes. *Serpine1* encodes the plasminogen activator inhibitor 1 (PAI-1), which modulates the extracellular matrix and inflammatory response, indirectly affecting adipocyte function. *Ucp1* is a marker of mature adipocytes in brown adipose tissue and contributes to thermogenesis, influencing lipid metabolism and energy balance, indicating cell maturity.

#### 3.4.3. Metabolism of Proteins in White Adipose Tissue

The protein metabolism pathway (Figure 9), identified by the R-MMU-392499 entry in Reactome, comprises eight secondary pathways, and the analyzed transcripts *Apoa1, Apoa5*, and *Ghrl* participate in three of these pathways. Apoa1 and Apoa5 participate in the post-translational protein modification and regulation of insulin-like growth factor (IGF) transport and uptake by IGF binding proteins (IGFBPs), both related to protein regulation through the serine/threonine kinase FAM20C, which phosphorylates secreted proteins and modulates their activity through interaction with the pseudokinase FAM20A.

In turn, *Ghrl* participates in the peptide hormone metabolism pathway through the synthesis, secretion, and deacylation of ghrelin, which is activated in its octanoylated form and binds to the GHS-R1a receptor in the hypothalamus, pituitary gland, and other tissues. In addition, insulin, which participates in the regulation of IGF transport and uptake pathway, inhibits ghrelin secretion. This reflects an interaction between ghrelin regulation and insulin metabolic pathways, influencing energy balance and overall metabolism. In the bloodstream, acylated ghrelin is deacetylated by enzymes such as butyrylcholinesterase. Downregulated transcripts by individual exposure to HFD and PM_2.5_ decrease the systemic activity of these metabolic processes in WAT.

#### 3.4.4. Cellular Responses to Stress, Cellular Responses to Stimuli, and Cellular Response to Chemical Stress in Heart Tissue

The analysis of the cellular responses to stress, cellular responses to stimuli, and cellular response to chemical stress pathways (Figure 10), cataloged in Reactome with the codes R-MMU-2262752, R-MMU-8953897, and R-MMU-9711123, revealed significant participation of three transcripts: *Gpx1*, *Pparα*, and *Sod1*. The cellular response to stimuli pathway is divided into two secondary pathways, one of which is the cellular stress response, which in turn is broken down into nine tertiary pathways, including the cellular response to chemical stress, divided into three sub-pathways. The transcribers analyzed participate in two sub-pathways: detoxification of ROS and cytoprotection by HMOX1. The identified mechanisms of involvement include response to elevated platelet cytosolic Ca^2+^, regulation of lipid metabolism by PPARα, generic transcription pathway, SUMOylation of intracellular receptors, GPX1 tetramer, Pparα:Rxrα: corepressors, and Pparα.

The response to elevated platelet cytosolic Ca^2+^ consists of activating phospholipase C enzymes, which trigger the generation of second messengers in the phosphatidylinositol pathway. This pathway increases intracellular calcium levels and activates protein kinase C (PKC). Phospholipase C hydrolyzes the phosphodiester bond in PIP2, forming 1,2-diacylglycerol (DAG) and 1,4,5-inositol trisphosphate (IP3). IP3 opens the Ca^2+^ channels in the dense tubular platelet system by raising intracellular Ca^2+^ levels. The DAG acts as a second messenger that regulates a family of Ser/Thr kinases, including PKC isoenzymes, increasing their affinity for phospholipids. In addition, some PKC isoenzymes are calcium-dependent, so their activation is enhanced by increased intracellular Ca^2+^. Platelets contain various PKC isoforms that DAG and/or Ca^2+^ can activate. The decrease in this biological process, due to the negative regulation of the transcripts Gpx1, Pparα, and Sod1 due to exposure to HFD and the combination of HFD and PM_2.5_, contributes to coagulation problems in the cardiovascular system.

The regulation of lipid metabolism by PPARα is mediated by this nuclear receptor type II, which forms heterodimers with RXRα, another nuclear receptor type II. The activation of PPARα occurs by binding ligands of fatty acids, particularly polyunsaturated fatty acids. This activation is essential for regulating lipid metabolism and cardiovascular function, as it reduces LDL and increases HDL, regulating lipid profiles. In addition, it is a target of fibrates, pharmacological agents used to treat dyslipidemias. Downregulation of *Gpx1*, *Pparα*, and *Sod1* transcripts by HFD exposure and mixed HFD and PM_2.5_ exposure affects this pathway, increasing the risk of dyslipidemia.

The generic transcription pathway, responsible for the differential regulation of gene transcription in eukaryotes, provides the general principles and mechanisms for this specific regulation in cells or tissues. In the cardiac context, RNA polymerase II is essential, as it regulates the transcription of protein-coding genes important for heart function. The downregulation of *Gpx1*, *Pparα*, and *Sod1* by exposure to HFD and the combination of HFD and PM_2.5_ affects this pathway, favoring cardiac dysfunctions.

SUMOylation of intracellular receptors is the process by which a SUMO (small ubiquitin-like modifier) group is added to nuclear receptors. Generally, this process induces transcriptional repression, which various mechanisms can carry out: interference with DNA binding, recruitment and retention of co-repressors in non-target gene promoters, relocation of nuclear receptors, interference with receptor dimerization, and interaction with other post-translational modifications. In the context of the cardiovascular system, SUMOylation of nuclear receptors is involved in regulating inflammation. The downregulation of *Gpx1*, *Pparα*, and *Sod1* by exposure to HFD and PM_2.5_ contributes to developing inflammatory processes in cardiac tissue.

GPX1 tetramer is the tetrameric form of the enzyme glutathione peroxidase 1 (GPX1). It is essential for cellular protection against oxidative stress. It reduces hydrogen peroxide (H_2_O_2_) and other hydroperoxides using glutathione as a cofactor, helping maintain the cell’s redox balance. Due to exposure to HFD and the combination of HFD and PM_2.5_, the downregulation of transcripts that induce this pathway can lead to a chronic oxidative environment and oxidative damage to cardiac tissue myocytes.

PPARα:RXRα: corepressors are activated in the absence of PPARα-activating ligands, recruiting co-repressors such as NCoR1, NCoR2, and histone deacetylases, which keeps chromatin in an inactive conformation and inhibits gene transcription. In contrast, the Pparα:Rxrα coactivator complex promotes gene transcription. The downregulation of the transcripts analyzed in the heart affects these pathways, causing dysregulation of crucial genes.

#### 3.4.5. Burn-Wound Healing in Cardiac Tissue

The burn-wound healing pathway, with identification WP5056 in WikiPathway (Figure 11), is based on a systematic and documented review in WikiPathway. Described initially as burn repair, it also has implications for cardiac repair after injury or stress. In this context, the genes *Col1a1*, *Tgfb1*, and *Tnf* participate in the formation of the extracellular matrix, the activation of fibroblasts, and the inflammatory response. Upregulation of these genes by exposure to PM_2.5_ could compromise the tissue’s structural integrity, affect healing, and decrease the effectiveness of the inflammatory response, resulting in insufficient cardiac repair and potential functional impairment of the myocardium.

#### 3.4.6. PPAR Signaling Pathway in Liver Tissue

The PPAR signaling pathway (Figure 12), identified with the mmu03320 input in KEGG, showed significant results, with transcripts affected by individual and combined exposure to HFD and PM_2.5_ in liver tissue. This pathway is mediated by three subtypes of PPAR receptors: PPARα, PPARβδ, and PPARγ, which present different expression patterns in vertebrates. Each of them, encoded by a distinct gene, has specific functions: PPARα regulates the elimination of circulating or cellular lipids through gene expression involved in lipid metabolism in the liver and skeletal muscle, PPARβδ participates in lipid oxidation and cell proliferation, and PPARγ promotes adipocyte differentiation, improving blood glucose uptake.

The analyzed transcripts *Acox1*, *Acs14*, *Pparα,* and *Scd1* showed a significant dysregulation in their expression after exposure to HFD and PM_2.5_, either individually or in combination, except *Acsl4*, which did not present statistically significant changes after exposure to HFD. These transcripts, partial mediators of this pathway through PPARα and PPARβδ receptors, alter fatty acid degradation and oxidation, bile acid biosynthesis, glycerophospholipid metabolism, and fatty acid transport in the liver.

#### 3.4.7. AMPK Signaling Pathway in Liver Tissue

The AMPK signaling pathway (Figure 13), with the KEGG registration mmu04152, presented significant results, with transcripts affected by individual and combined exposure to HFD and PM_2.5_ in liver tissue. It is a pathway where the AMPK, an evolutionarily conserved serine–threonine kinase, is a sensor of cellular energy status. AMPK is activated by an increase in the proportion of AMP in the cell caused by metabolic stress conditions that interfere with ATP production, such as fasting and hypoxia, which increases ATP consumption and muscle contraction. Several regulatory kinases, including liver kinase B1, calcium/calmodulin beta kinase, and TGF-beta-activated kinase-1, can activate AMPK by phosphorylating a threonine residue at its alpha catalytic subunit. Once activated, AMPK inhibits energy-consuming biosynthetic pathways, such as protein, fatty acid, and glycogen synthesis, while activating ATP-producing catabolic pathways, such as fatty acid oxidation and glycolysis.

According to the results, the *Fasn*, *Scd1*, and *Sirt1* transcripts exhibited dysregulated expression after single and combined exposure to HFD and PM_2.5_. These transcripts partially participate in the AMPK signaling pathway, altering the activation of mitochondrial biogenesis, fatty acid biosynthesis, and unsaturated fatty acid biosynthesis in the liver.

#### 3.4.8. Non-Alcoholic Fatty Liver Disease

Non-alcoholic fatty liver disease (NAFLD) (Figure 14), identified with the entry mmu04932 in KEGG, showed significant results, with transcripts affected by single and combined exposure to HFD and PM_2.5_ in liver tissue. NAFLD covers a spectrum ranging from mild steatosis to non-alcoholic steatohepatitis (NASH), which is characterized by inflammation and liver fibrosis. NASH can progress to cirrhosis and hepatocellular carcinoma (HCC). Figure 14 illustrates the progression of NAFLD in its different stages. In the former, an excessive accumulation of lipids is observed, mainly due to the induction of insulin resistance, which prevents insulin’s adequate suppression of FFA. In addition, two TFs, SREBP-1c and PPARα, activate key lipogenesis enzymes and increase FAA synthesis in the liver. In the second stage, ROS production is increased due to oxidative stress through mitochondrial fatty acid beta-oxidation and endoplasmic reticulum (ER) stress, which leads to lipid peroxidation that can cause the production of cytokines (Fas ligand, TNF-alpha, IL-8, and TGF), promoting cell death, inflammation, and fibrosis, favoring the production of cytokines and the initiation of HCC.

The results revealed a dysregulation in the expression of the transcripts *Fas*, *Pparα*, and *Tnf* after the individual and combined exposure to HFD and PM_2.5_. These transcripts participate partially in the three stages of this pathway. In the first stage, TNFα and PPAR signaling is affected, reducing fatty acid oxidation and increasing glucose levels, which leads to the second stage. In the third stage, the disruption of FAS receptors indirectly contributes to hepatocyte apoptosis.

## 4. Discussion

The present systematic review and meta-analysis aimed to form a comprehensive overview of the molecular mechanisms underlying metabolically abnormal obesity induced by an HFD and exposure to PM_2.5_. Thirty-three original articles of moderate to high quality, published in journals indexed on Web of Science, Scopus, and PubMed, were analyzed to test the hypothesis that chronic exposure to an HFD and PM_2.5_ promotes a chronically oxidative cellular environment, which dysregulates the transcription of genes related to small molecule transport, PPAR and HIF-1 signaling pathways, cytokine–cytokine receptor interaction, and adipogenesis gene pathways through oxidative modification of the transcription factors and enzymes involved in DNA methylation. Our results indicate that combining an HFD and prolonged exposure to PM_2.5_ produces a more significant increase in body weight than individual exposure to PM_2.5_ or HFD. In contrast, Goettems-Fiorin et al. [143] observed an increase in weight in murine exposed to HFD and PM_2.5_. Still, they did not find statistically significant differences between the control group and those exposed individually or combined to both factors.

Similarly, the review by Guardia and Wang [144] reports that, in animals exposed to PM_2.5_, WAT hypertrophy occurred independently of changes in energy intake. However, this review also mentions that mice exposed to PM_2.5_ in utero show microglial activation, increased anxiety, and higher body weight in adulthood compared to the control group, suggesting that prolonged exposure to PM_2.5_ could remodel the circuits that regulate the feeding behavior and energy balance. The inconsistency between these results could be due to the high heterogeneity observed in our meta-analysis, both globally and in subgroups, probably related to phenotypic differences in the size of the murine species studied and variations in the design and scope of the investigations.

Based on the oxidative stress biomarkers analyzed, only SOD showed statistically significant results, although with high heterogeneity. SOD is an essential antioxidant enzyme that protects cells from oxidative damage by catalyzing the conversion of superoxide radicals to oxygen and hydrogen peroxide [145,146]. An increase in SOD levels usually reflects an adaptive response of the organism to an increase in oxidative stress [147,148].

The analysis of the data shows that the combined exposure to HFD and PM_2.5_ caused a more notable increase in SOD levels compared to the individual exposures, which recorded similar increases between them. This observation suggests that simultaneous exposure to both factors enhances oxidative damage to a greater extent than exposure to each separately, consistent with our hypothesis that chronic exposure to HFD and PM_2.5_ promotes a chronically oxidative cellular environment. The elevation of SOD in this context can be interpreted as an indicator of the organism’s adaptive response to a state of exacerbated oxidative stress. This result coincides with a recent review on the interaction of environmental factors in the metabolic processes of metastasis [149], suggesting that the combination of HFD and PM_2.5_ not only alters the antioxidant response but also contributes to greater cellular vulnerability to oxidative damage. The high variability across the studies analyzed may stem from differences in exposure doses and application frequency, underscoring the need for standardized protocols to accurately assess the effects of these exposures in the context of oxidative stress.

Among the metabolic biomarkers associated with increased body weight, individual exposure to HFD and PM_2.5_ was observed to cause significant increases in four parameters: total cholesterol, HDL, insulin, and glucose, with no or low heterogeneity. These results could suggest a mechanistic interaction [150] between both factors, consistent with our previous study [20]. On the other hand, combined exposure to HFD and PM_2.5_ also led to significant increases in HDL and insulin, with no or low heterogeneity. This finding is relevant since it suggests a biological interaction [150], indicating a possible synergy in its metabolic impact. Our results align with other recent reviews [144,151]. The consistent response in HDL and insulin, even under simultaneous exposure to both factors, underscores the importance of jointly evaluating these exposures when assessing the metabolic risk.

When the transcripts were examined with the biological pathway enrichment analysis, we found that only clusters of three or four genes showed statistically significant results for each molecular pathway in BAT, WAT, heart, and liver tissue. This result supports our hypothesis that chronic exposure to HFD and PM_2.5_ dysregulates the transcription of genes related to various metabolic pathways. Identifying specific gene clusters with statistical significance suggests that exposure to these environmental factors has a selective impact on regulating particular biological pathways, which could be fundamental to understanding how the interaction between HFD and PM_2.5_ alters cellular function and contributes to metabolic alterations.

In BAT, PM_2.5_ exposure dysregulated *Cebpa*, *Pparα*, *Serpine1*, and *Ucp1* transcripts, affecting genes related to adipogenesis, which is consistent with our previous findings [20]. On the other hand, HFD altered the regulation of *Tnf*, *Lepr*, and *Pparα*, impacting the adipocytokine signaling pathway, which coincides with the reports of Dogan and Brockmann [152], who reported that *Pparα*, along with other downregulated genes, is involved in inflammatory pathways, particularly in the adipocytokine signaling pathway and in the complement and coagulation cascades in epididymal adipose tissue. These results partially support our hypothesis that chronic exposure to HFD and PM_2.5_ deregulates the transcription of genes related to specific biological pathways. Although no evidence was found for some of the hypothesized pathways, such as small-molecule transport and HIF-1 signaling, identifying pathways related to adipogenesis and cytokine regulation is relevant, suggesting that HFD and PM_2.5_ interact by affecting complementary pathways [150]. While HFD promotes a pro-inflammatory environment through the adipocytokine signaling pathway, exposure to PM_2.5_ interferes with the ability of BAT to carry out adipogenesis and maintain thermogenesis, exacerbating metabolic dysfunction in this tissue.

In WAT, HFD and PM_2.5_ exposure affected the protein metabolism pathway, downregulating *Apoa1*, *Apoa5*, and *Ghrl* transcripts, suggesting an impact on lipid metabolism. However, unlike our initial hypothesis, no significant alterations were observed in the inferred biological pathways. This finding differs from Guerra-Cantera et al. [153], who observed more significant alterations in this pathway after a low-fat diet. Likewise, minimal changes were reported in the same pathway [154]. These discrepancies could be due to differences in the time of exposure to HFD and the tissues analyzed since neither study focused on WAT. The concurrent alteration of this pathway by both exposures suggests a mechanistic interaction [150], possibly amplifying the negative impact on WAT, which could compromise its metabolic and regulatory function.

Individual and combined exposure to HFD and PM_2.5_ in liver tissue affects the same pathways. The AMPK signaling pathway showed deregulation of *Fasn*, *Scd1*, and *Sirt1* transcripts, consistent with other authors [155,156,157] who independently observed similar effects in murine models exposed to HFD or PM_2.5_. The affectation of these pathways suggests a possible interaction [150] between HFD and PM_2.5_, intensifying transcriptomic alterations in the liver and potentially aggravating metabolic dysfunctions.

The non-alcoholic fatty liver disease showed a transcriptional alteration in the *Fas*, *Pparα*, and *Tnf* genes, consistent with literature where similar changes induced by HDF were observed [158,159]. In addition, a review reports alterations in these transcripts after exposure to PM_2.5_ [160]. The concurrent alteration of these genes is a good proxy of the interaction [150] between HFD and PM_2.5_, which could synergistically affect transcriptional dysregulation, favoring the development, progression, and complication of non-alcoholic fatty liver disease.

In the liver, the PPAR signaling pathway showed deregulation in Acox1, Pparα, and Scd1 transcripts after exposure to HFD, in agreement with Tu et al. [161]. Consistent with our initial hypothesis, the transcription of *Acox1*, *Acs14*, *Pparα*, and *Scd1* was altered by PM_2.5_ [162,163]. The identified pooled effect possibly indicates a mechanistic and biological interaction [150], which enhances the PPAR signaling pathway’s impact and aggravates liver dysfunctions associated with obesity and exposure to PM_2.5_.

In cardiac tissue, exposure to HFD combined with PM_2.5_ activates pathways related to cellular responses to stress, cellular responses to stimuli, and cellular response to chemical stress, with a downregulation of *Apoa1*, *Apoa5*, and *Ghrl* transcripts, which does not agree with our initial hypothesis. Similar alterations in pulmonary endothelial cells exposed to PM_2.5_ and effects in adipocytes after HFD were described in Almeida-Silva et al. [164] and Jarc and Petan [165], respectively.

A HFD induces the accumulation of lipid droplets in myocytes, favoring the development of heart disease [166,167]. In this context, lipid droplets could modulate the cellular stress response, integrating inflammatory and metabolic processes that affect immune cells and various tissues [165]. This enhances biological interactions that amplify cellular and metabolic stress, exacerbating dysfunction in cardiac tissue and potentially contributing to the development of heart disease.

A limitation of the performed meta-analysis is the moderate to high heterogeneity observed in some studies, reflected in the *I*^2^ index values. Although subgroup analyses were applied to address these differences, variability in biological models, the tissues analyzed, and measurement methods may have contributed to this heterogeneity. While this diversity in experimental conditions can enrich our understanding of the effects of HFD and PM_2.5_ in different contexts, it also somewhat restricts the ability to generalize the findings to other settings. However, the statistical approaches used to handle heterogeneity allow conclusions to be robust within the limits of the included studies.

## 5. Prospectives

The findings from this systematic review and meta-analysis offer directions for future research that could enhance the understanding of the interaction between HFD and PM_2.5_ in metabolically abnormal obesity. Epidemiological studies in human populations are needed to assess chronic co-exposure to both factors and generate evidence regarding the combined effects of these exposures. Additionally, longitudinal studies could be valuable for clarifying the temporal progression of metabolic alterations associated with obesity, thereby improving the understanding of underlying causal relationships.

Investigating tissue-specific responses to combined HFD and PM_2.5_ exposure would be valuable in the experimental domain. This would provide a more detailed understanding of molecular alterations in critical tissues such as adipose, cardiac, and hepatic tissues and could facilitate the exploration of signaling pathways identified as potential therapeutic targets.

Furthermore, integrating bioinformatics tools and multi-omic analysis could provide promising approaches to advancing the understanding of molecular interactions between dietary and environmental exposures. This would allow for a better control of confounding variables and increase the results’ precision.

## 6. Conclusions

This systematic review and meta-analysis demonstrate that the combined exposure to an HFD and PM_2.5_ significantly dysregulates the expression of genes associated with signaling pathways and metabolism, resulting in increased body weight, oxidative stress, and elevated levels of HDL and insulin. The interaction between these factors alters key biological pathways, contributing to metabolic dysfunction in various tissues, such as the liver, adipose tissue, and the cardiovascular system. The results suggest a mechanistic synergy between HFD and PM_2.5_, which may contribute to developing conditions such as NAFLD and other disorders linked to metabolically abnormal obesity.

It is essential to highlight that the PPAR signaling pathway, identified as significantly affected by HFD and PM_2.5_, plays a crucial role in regulating metabolic processes and may represent a key target in therapeutic strategies. Although the participation of the HIF-1 signaling pathway was not confirmed in this study, the identified pathways, such as those related to oxidative stress, adipogenesis, and cytokine interactions, are consistent with the metabolic alterations commonly associated with obesity and its complications.

Exposure to atmospheric pollution and obesogenic environments is widespread globally, underscoring the urgent need to address both factors in public health strategies. Implementing interventions that reduce air pollution and promote healthy diets could be vital in mitigating the risks of chronic metabolic diseases. These strategies must focus mainly on the most vulnerable populations, who face more significant risks associated with these combined exposures.

Finally, the identification of molecular pathways altered by exposure to a high-fat diet and PM_2.5_ particles opens new possibilities for developing therapeutic approaches targeting the implicated molecular mechanisms. This could be crucial for improving the prevention and treatment of metabolic diseases, contributing to better-managing obesity and its associated complications.

## Figures and Tables

**Figure 1 biomolecules-14-01607-f001:**
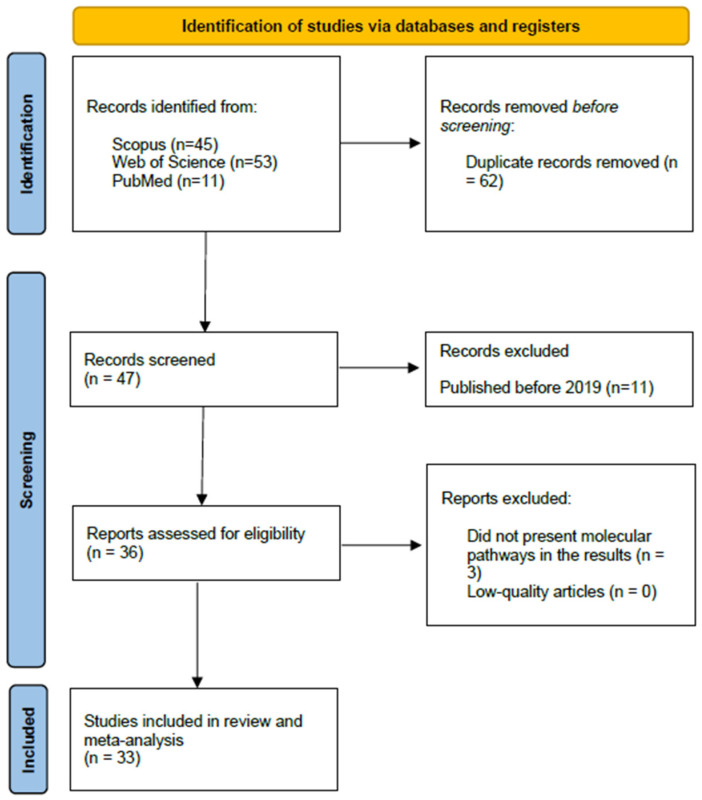
Flow diagram for study selection.

**Figure 2 biomolecules-14-01607-f002:**
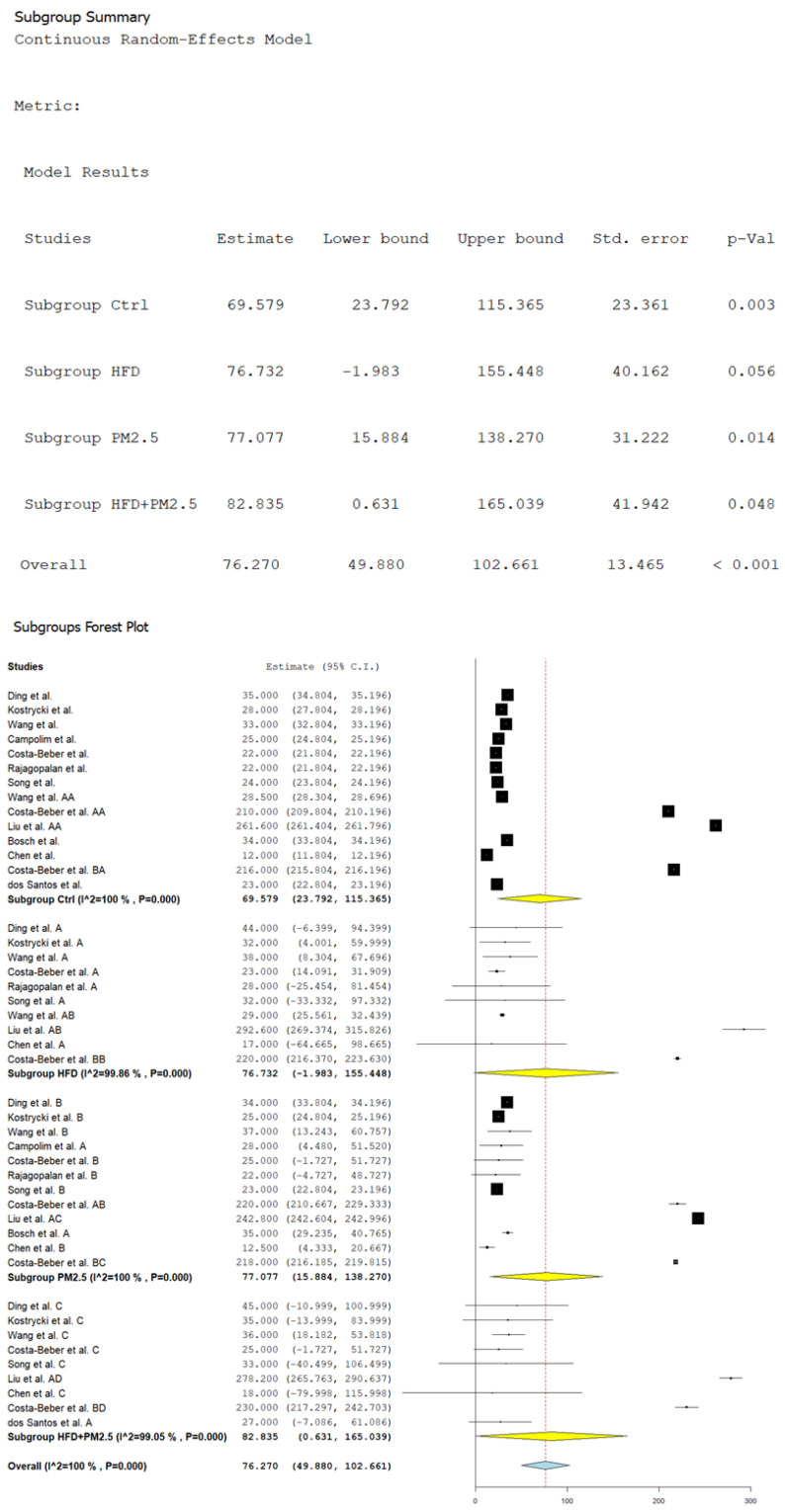
Random-effects model and subgroup forest plot of body-weight gain by exposure to HFD, PM_2.5_, and their combination in mice studies. Ding et al., Ding et al. A, Ding et al. B, and Ding et al. C = reference [111]. Kostrycki et al., Kostrycki et al. A, Kostrycki et al. B, and Kostrycki et al. C = reference [112]. Wang et al., Wang et al. A, Wang et al. B, and Wang et al. C = reference [113]. Campolim et al., and Campolim et al. A = reference [115]. Costa-Beber et al., Costa-Beber et al. A, Costa-Beber et al. B, and Costa-Beber et al. C = reference [116]. Rajagopalan et al., Rajagopalan et al. A, and Rajagopalan et al., B = reference [120]. Song et al., Song et al. A, Song et al. B, and Song et al. C = reference [121]. Wang et al. AA, and Wang et al. AB = reference [122]. Costa-Beber et al. AA, and Costa-Beber et al. AB = reference [123]. Liu et al. AA, Liu et al. AB, Liu et al. AC, and Liu et al. AD = reference [125]. Bosch et al., and Bosch et al. A = reference [131]. Chen et al., Chen et al. A, Chen et al. B, and Chen et al. C = reference [132]. Costa-Beber et al. BA, Costa-Beber et al. BB, Costa-Beber et al. BC, and Costa-Beber et al. BD = reference [133]. dos Santos et al., and dos Santos et al. A = reference [135]. Black squares represent the estimated effect size (mean difference) of each individual study, with the size of the square being proportional to the weight of the study in the combined estimate. Black vertical lines indicate the null or no-effect value, which corresponds to 0 for a mean difference analysis, representing no difference between groups. The dotted vertical line represents the overall combined effect size estimate, reflecting the central value of the combined effect across all studies. Horizontal lines represent the confidence intervals (CI) of the estimated effect size for each study, showing the range within which the true effect size is expected to lie with 95% confidence. The length of the line indicates the precision of the estimate. The yellow diamond represents the combined mean difference estimates for each subgroup within the meta-analysis, showing the effect size and its corresponding 95% CI. The blue diamond represents the overall combined effect size across all studies, integrating the results of all subgroups, with its corresponding 95% CI.

**Figure 3 biomolecules-14-01607-f003:**
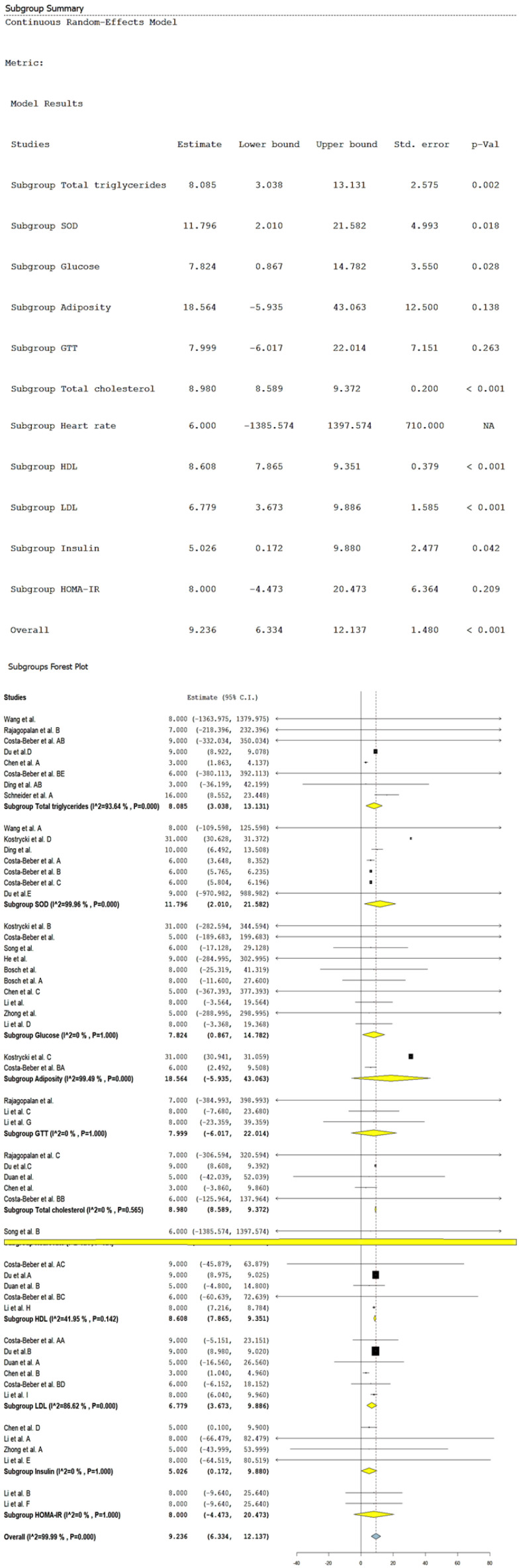
Random-effects model and subgroup forest plot of metabolic biomarkers induced by HFD. NA = Not applicable. Wang et al., and Wang et al. A = reference [113]. Rajagopalan et al., Rajagopalan et al. B, and Rajagopalan et al. C = reference [120]. Costa-Beber et al. AA, Costa-Beber et al. AB, Costa-Beber et al. B, and Costa-Beber et al. AC = [123]. Du et al. A, Du et al. B, Du et al. C, Du et al. D, and Du et al. E = reference [126]. Chen et al., Chen et al. A, Chen et al. B, Chen et al. C, and Chen et al. D = reference [132]. Costa-Beber et al. BB, Costa-Beber et al. BC, Costa-Beber et al. BD, and Costa-Beber et al. BE = reference [133]. Ding et al., and Ding et al. AB = reference [134]. Schneider et al. A = reference [138]. Kostrycki et al. B, Kostrycki et al. C, and Kostrycki et al. D = reference [112]. Costa-Beber et al., Costa-Beber et al. A, and Costa-Beber et al. BA = reference [116]. Song et al., and Song et al. B = reference [121]. He et al. = reference [129]. Bosch et al., and Bosch et al. A = reference [131]. Li et al., Li et al. A, Li et al. C, Li et al. G, Li et al. H, and Li et al. I = reference [137]. Zhong et al., and Zhong et al. A = reference [141]. Li et al. B, Li et al. D, Li et al. E, and Li et al. F = reference [142]. Duan et al., Duan et al. A, and Duan et al. B = reference [127]. Black squares represent the estimated effect size (mean difference) of each individual study, with the size of the square being proportional to the weight of the study in the combined estimate. Black vertical lines indicate the null or no-effect value, which corresponds to 0 for a mean difference analysis, representing no difference between groups. The dotted vertical line represents the overall combined effect size estimate, reflecting the central value of the combined effect across all studies. Horizontal lines represent the confidence intervals (CI) of the estimated effect size for each study, showing the range within which the true effect size is expected to lie with 95% confidence. The length of the line indicates the precision of the estimate. The yellow diamond represents the combined mean difference estimates for each subgroup within the meta-analysis, showing the effect size and its corresponding 95% CI. The blue diamond represents the overall combined effect size across all studies, integrating the results of all subgroups, with its corresponding 95% CI.

**Figure 4 biomolecules-14-01607-f004:**
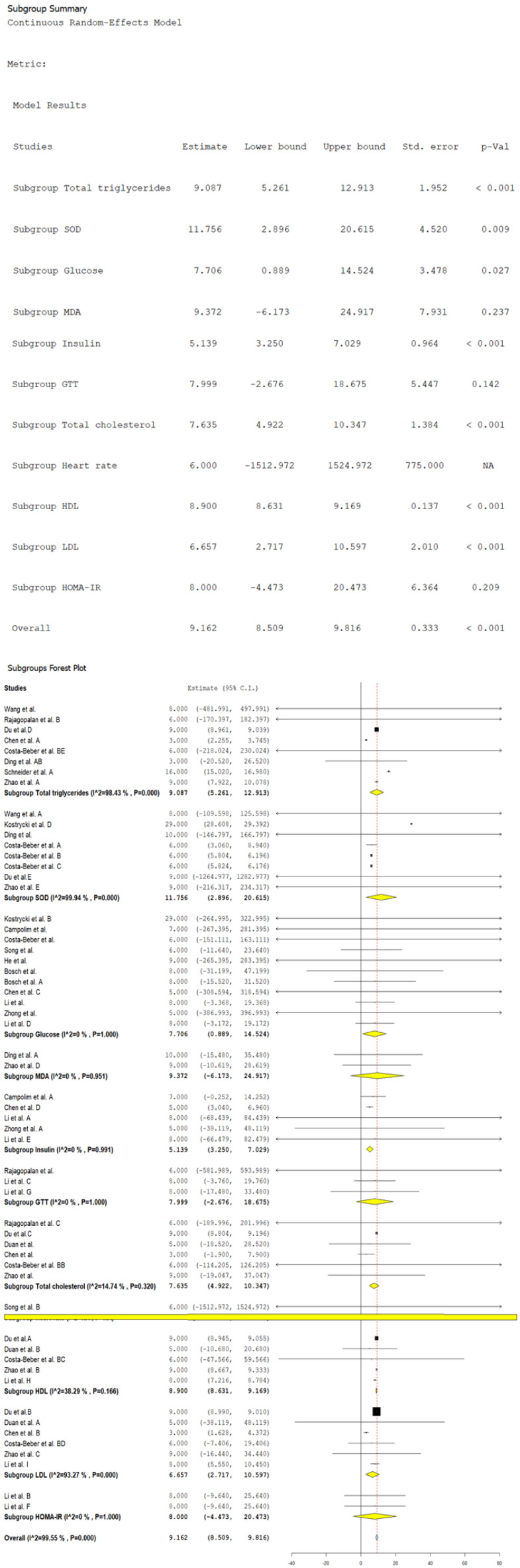
Random-effects model and subgroup forest plot of metabolic biomarkers induced by PM_2.5._ NA = Not applicable. Wang et al., and Wang et al. A = reference [113]. Rajagopalan et al., Rajagopalan et al. B, and Rajagopalan et al. C = reference [120]. Du et al. A, Du et al. B, Du et al. C, Du et al. D, and Du et al. E = reference [126]. Chen et al., Chen et al. A, Chen et al. B, Chen et al. C, and Chen et al. D = reference [132]. Costa-Beber et al. C, Costa-Beber et al. BB, Costa-Beber et al. BC, Costa-Beber et al. BD, and Costa-Beber et al. BE = reference [133]. Ding et al. A, and Ding et al. AB = reference [134]. Schneider et al. A = reference [138]. Zhao et al., Zhao et al. A, Zhao et al. B, Zhao et al. C, Zhao et al. D, and Zhao et al. E = reference [140]. Kostrycki et al. B, and Kostrycki et al. D = reference [112]. Ding et al. = reference [111]. Costa-Beber et al., and Costa-Beber et al. A = reference [116]. Costa-Beber et al. B = reference [123]. Campolim et al., and Campolim et al. A = reference [115]. Song et al., and Song et al. B = reference [121]. He et al. = reference [129]. Bosch et al., and Bosch et al. A = reference [131]. Li et al., Li et al. A, Li et al. B, Li et al. C, Li et al. D, Li et al. E, Li et al. F, Li et al. G, Li et al. H, and Li et al. I = reference [137]. Zhong et al., and Zhong et al. A = reference [141]. Duan et al., Duan et al. A, and Duan et al. B = reference [127]. Black squares represent the estimated effect size (mean difference) of each individual study, with the size of the square being proportional to the weight of the study in the combined estimate. Black vertical lines indicate the null or no-effect value, which corresponds to 0 for a mean difference analysis, representing no difference between groups. The dotted vertical line represents the overall combined effect size estimate, reflecting the central value of the combined effect across all studies. Horizontal lines represent the confidence intervals (CI) of the estimated effect size for each study, showing the range within which the true effect size is expected to lie with 95% confidence. The length of the line indicates the precision of the estimate. The yellow diamond represents the combined mean difference estimates for each subgroup within the meta-analysis, showing the effect size and its corresponding 95% CI. The blue diamond represents the overall combined effect size across all studies, integrating the results of all subgroups with its corresponding 95% CI.

**Figure 5 biomolecules-14-01607-f005:**
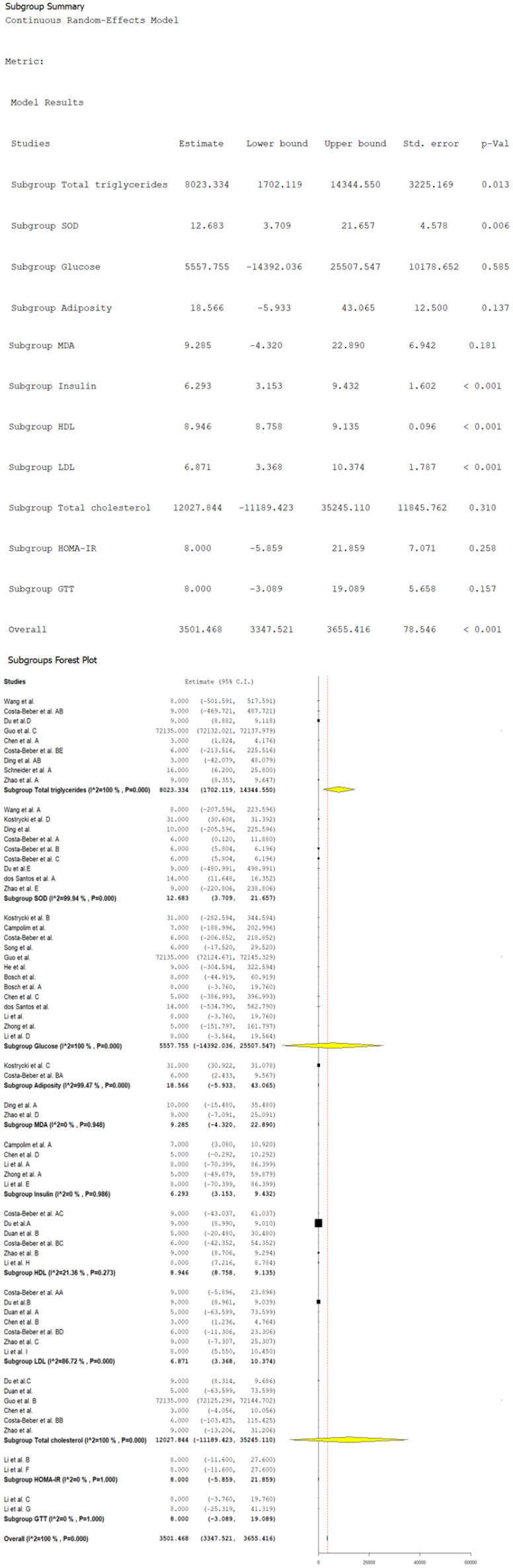
Random-effects model and subgroup forest plot of metabolic biomarkers induced by HFD and PM_2.5_. Wang et al., and Wang et al. A = reference [113]. Costa-Beber et al., Costa-Beber et al. A, Costa-Beber et al. AA, Costa-Beber et al. AB, Costa-Beber et al. AC, and Costa-Beber et al. BA = reference [116]. Du et al. A, Du et al. B, Du et al. C, Du et al. D, and Du et al. E = reference [126]. Guo et al., Guo et al. B, and Guo et al. C = reference [128]. Chen et al., Chen et al. A, Chen et al. B, Chen et al. C, and Chen et al. D = reference [132]. Costa-Beber et al. BB, Costa-Beber et al. BC, Costa-Beber et al. BD, Costa-Beber et al. BE, and Costa-Beber et al. C = reference [133]. Ding et al., Ding et al. A, and Ding et al. AB = reference [134]. Schneider et al. A = reference [138]. Zhao et al., Zhao et al. A, Zhao et al. B, Zhao et al. C, Zhao et al. D, and Zhao et al. E = reference [140]. Kostrycki et al. B, Kostrycki et al. C, and Kostrycki et al. D = reference [112]. Costa-Beber et al. B = reference [123]. Santos et al., and dos Santos et al. A = reference [135]. Campolim et al., Campolim et al. A = reference [115]. Song et al. = reference [121]. He et al. = reference [129]. Bosch et al., and Bosch et al. A = reference [131]. Li et al., Li et al. A, Li et al. B, Li et al. C, Li et al. D, Li et al. E, Li et al. F, Li et al. G, Li et al. H, and Li et al. I = reference [137]. Zhong et al., and Zhong et al. A = reference [141]. Black squares represent the estimated effect size (mean difference) of each individual study, with the size of the square being proportional to the weight of the study in the combined estimate. Black vertical lines indicate the null or no-effect value, which corresponds to 0 for a mean difference analysis, representing no difference between groups. The dotted vertical line represents the overall combined effect size estimate, reflecting the central value of the combined effect across all studies. Horizontal lines represent the confidence intervals (CI) of the estimated effect size for each study, showing the range within which the true effect size is expected to lie with 95% confidence. The length of the line indicates the precision of the estimate. The yellow diamond represents the combined mean difference estimates for each subgroup within the meta-analysis, showing the effect size and its corresponding 95% CI. The blue diamond represents the overall combined effect size across all studies, integrating the results of all subgroups with its corresponding 95% CI.

**Figure 6 biomolecules-14-01607-f006:**
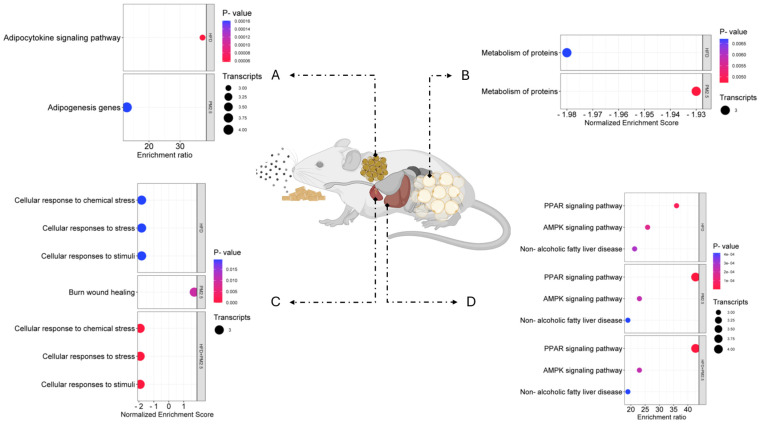
Gene set enrichment analysis (GSEA) and over-representation analysis (ORA) of biological pathways induced by HFD, PM_2.5_, and HFD + PM_2.5_ exposure in different tissues. (**A**) ORA analysis in BAT. (**B**) GSEA analysis in WAT. (**C**) GSEA analysis in cardiac tissue. (**D**) ORA analysis in hepatic tissue. Graphics were created in SRPlot (1 August 2024, https://www.bioinformatics.com.cn/en), and figures were designed using the BioRender program (15 August 2024, https://app.biorender.com/).

**Figure 7 biomolecules-14-01607-f007:**
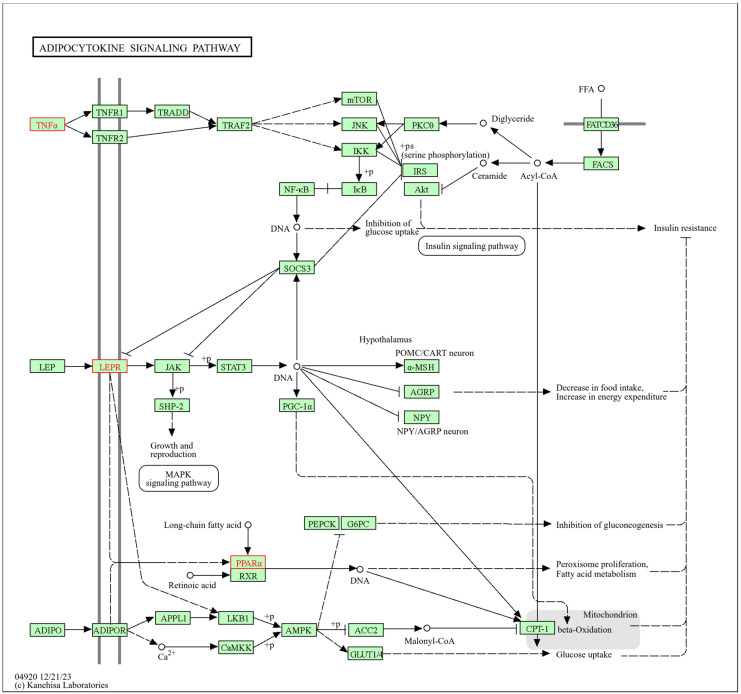
KEGG network diagram of the adipocytokine signaling pathway. White boxes: biological pathway maps. Green boxes: genes or gene products. Circles: molecules. Solid line arrows: direct relationships or molecular interactions. Dashed line arrows: indirect relationships or unknown reactions. Green boxes + arrows + circles + arrows = gene expression relationship. Red indicates differentially expressed transcripts after HFD exposure in brown adipose tissue (BAT).

**Figure 8 biomolecules-14-01607-f008:**
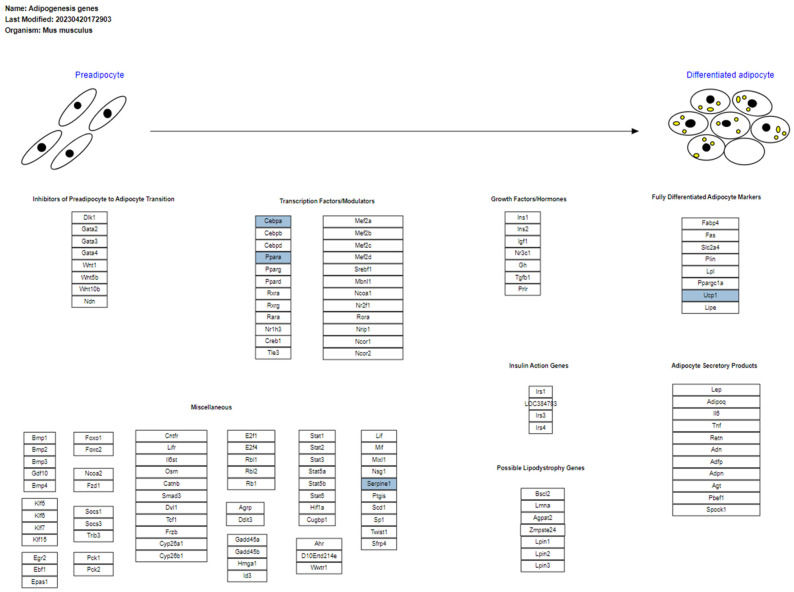
WikiPathways diagram of adipogenesis genes. The differentially expressed transcripts after PM_2.5_ exposure in brown adipose tissue (BAT) are in blue.

**Figure 9 biomolecules-14-01607-f009:**
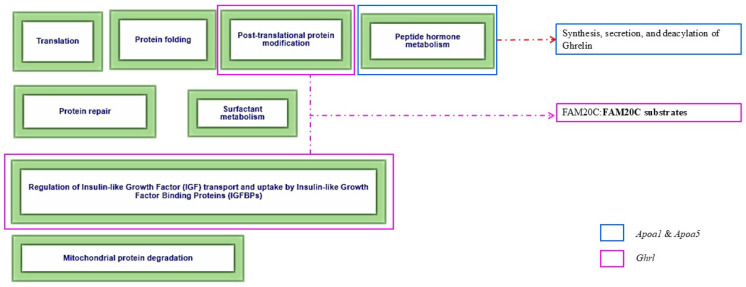
Reactome diagram of protein metabolism. In violet and blue, biological and transcript processes are differentially expressed after individual exposure to HFD and PM_2.5_ in white adipose tissue (WAT).

**Figure 10 biomolecules-14-01607-f010:**
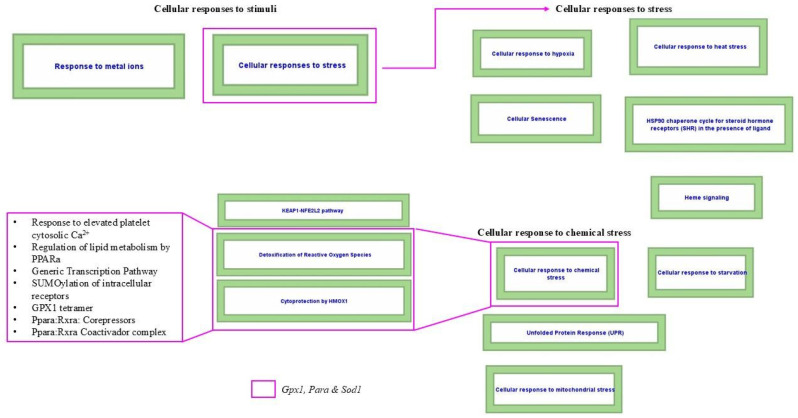
Reactome diagram of the cellular responses to stress, cellular responses to stimuli, and cellular response to chemical stress metabolism of protein pathways. In violet, differentially expressed biological and transcript processes after exposure to HFD and HFD + PM_2.5_ in cardiac tissue.

**Figure 11 biomolecules-14-01607-f011:**
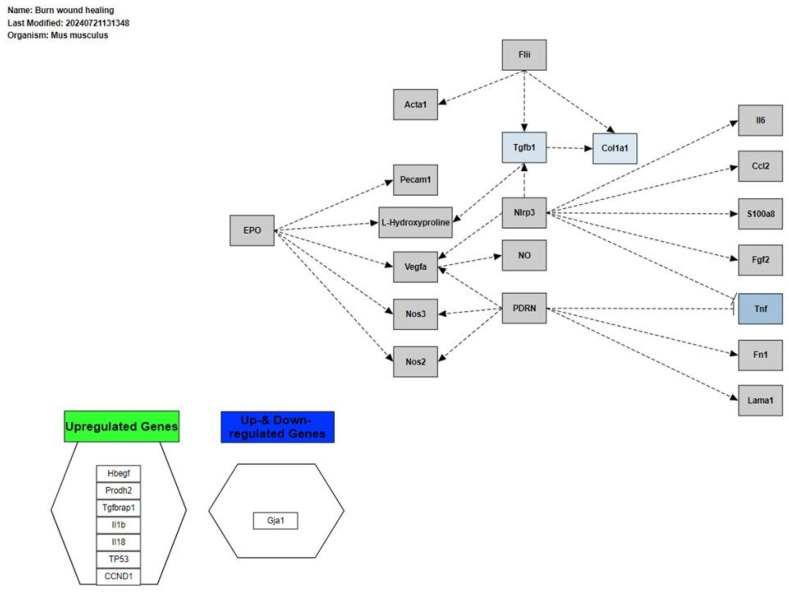
WikiPathway outline of burn-wound healing. Blue shading highlights differentially expressed transcripts in cardiac tissue following PM_2.5_ exposure. The intensity of the blue color indicates the extent of gene dysregulation, with darker shades representing higher levels of dysregulation within the pathway.

**Figure 12 biomolecules-14-01607-f012:**
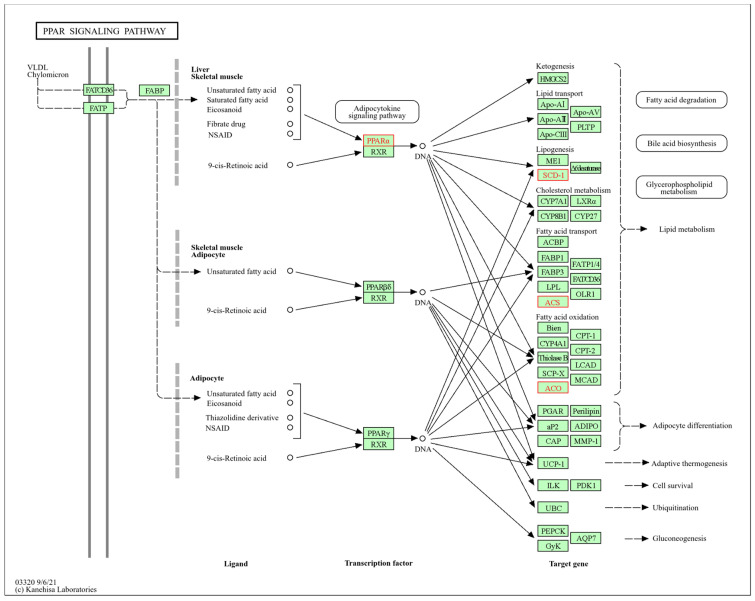
KEGG network diagram of the PPAR signaling pathway. White squares: maps of biological pathways. Green boxes: genes or gene products. Circles: molecules. Solid line arrows: direct relationship or molecular interaction. Dashed line arrows: indirect relationship or unknown reaction. Green squares + arrows + circles + arrows = gene expression ratio. In red, the transcripts that show a differential expression in liver tissue after individual and combined exposure to HFD and PM_2.5_ stand out, except *Acsl4*, which did not present statistically significant results after exposure to HFD.

**Figure 13 biomolecules-14-01607-f013:**
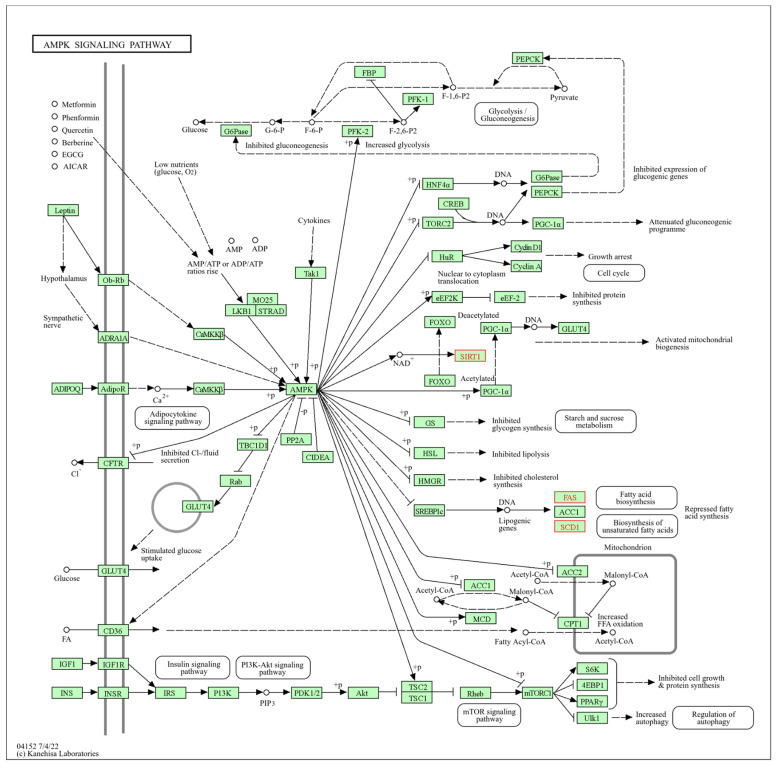
Diagram of red KEGG of AMPK signaling pathway. White squares: maps of biological pathways. Green boxes: genes or gene products. Circles: molecules. Solid line arrows: direct relationship or molecular interaction. Dashed line arrows: indirect relationship or unknown reaction. Green squares + arrows + circles + arrows = gene expression ratio. Highlighted in red are transcripts that show differential expression in liver tissue after individual and combined exposure to HFD and PM_2.5_. FAS is the protein encoded by *Fasn*.

**Figure 14 biomolecules-14-01607-f014:**
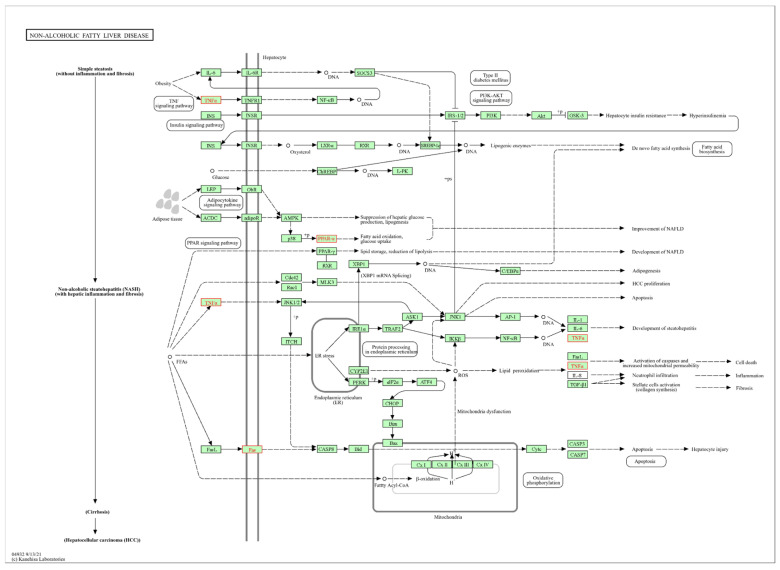
Diagram of red KEGG of Non-alcoholic fatty liver disease. White squares: maps of biological pathways. Green boxes: genes or gene products. Circles: molecules. Solid line arrows: direct relationship or molecular interaction. Dashed line arrows: indirect relationship or unknown reaction. Green squares + arrows + circles + arrows = gene expression ratio. Highlighted in red are transcripts that show differential expression in liver tissue after individual and combined exposure to HFD and PM_2.5_.

**Table 1 biomolecules-14-01607-t001:** Study characteristics of articles included in the systematic review and meta-analysis.

No.	Ref.	StD	PBM	Exposure	n	TCL	QS
1	[111]	CCS	C57BL/6J mice	PM_2.5_, HFD, and PM_2.5_ + HFD	40	Liver	8
2	[112]	L	Isogenic B6129F2/J mice	PM_2.5_, HFD, and PM_2.5_ + HFD	60	Blood	8
3	[113]	CCS	apoE^−/−^ mice	PM_2.5_, HFD, and PM_2.5_ + HFD	32	Liver	8
4	[114]	CCS	C57BL/6 J or WT mice TLR4-deficient mice	HFD and PM_2.5_ + HFD	13	Blood	8
5	[115]	CCS	C57BL/6 J or WT mice TLR4-deficient mice	PM_2.5_	36	Hypothalamic	8
6	[116]	CCS	B6129SF2/J mice	PM_2.5_, HFD, and PM_2.5_ + HFD	23	Adipose tissue	9
7	[117]	CCS	U937-derived macrophages and human aortic endothelial cells	PM_2.5_, HFD, and PM_2.5_ + HFD	NA	U937 human cells and HAEC human cells	8
8	[118]	CCS	C57BL/6J mice	PM_2.5_, HFD, and PM_2.5_ + HFD	40	Heart	8
9	[119]	C	Humans	PM_2.5_	38,824	Adipose tissue	7
10	[120]	CCS	C57BL/6J mice	PM_2.5_ and HFD	36	Liver, heart and adipose tissue	8
11	[121]	CCS	C57BL/6 J mice	PM_2.5_, HFD, and PM_2.5_ + HFD	24	Heart, hypothalamic, and lung	8
12	[122]	CCS	ApoE^−/−^ C57BL/6 J	PM_2.5_	14	Vascular	9
13	[123]	CCS	Wistar rats	PM_2.5_, HFD, and PM_2.5_ + HFD	36	Vascular	8
14	[124]	CCS	C57BL/6 mice	PM_2.5_, HFD, and PM_2.5_ + HFD	40	Vascular	8
15	[125]	CCS	SD rats	PM_2.5_, HFD, and PM_2.5_ + HFD	36	Intestinal	8
16	[126]	CCS	C57BL/6J mice	PM_2.5_, HFD, and PM_2.5_ + HFD	40	Liver	8
17	[127]	CCS	apoE^−/−^ mice	PM_2.5_, HFD, and PM_2.5_ + HFD	40	Macrophages	8
18	[128]	T	Humans	PM_2.5_ + HFD	90,086	Blood	80%
19	[129]	CCS	C57BL/6 J mice	PM_2.5_, HFD, and PM_2.5_ + HFD	34	Spleen	8
20	[130]	CCS	C57BL/6 mice	PM_2.5_, HFD, and PM_2.5_ + HFD	50	Heart	8
21	[131]	CCS	C57BL/6N mice	PM_2.5_	54	Lung	8
22	[132]	CCS	C57BL/6J mice	PM_2.5_ and PM_2.5_ + HFD	120	Lung	8
23	[133]	CCS	Wistar rats	PM_2.5_, HFD, and PM_2.5_ + HFD	24	Cardiovascular	8
24	[134]	CCS	C57BL/6 J mice	PM_2.5_	40	Liver	8
25	[135]	CCS	B6129SF2/J mice	PM_2.5_, HFD, and PM_2.5_ + HFD	31	Muscle, gastrocnemius, soleus, pancreas, and adipose tissue	8
26	[136]	T	Humans	PM_2.5_	99,556	Blood	70%
27	[137]	CCS	Wistar rats	PM_2.5_, HFD, and PM_2.5_ + HFD	32	Blood	8
28	[138]	CCS	C57Bl/6 mice	PM_2.5_, HFD, and PM_2.5_ + HFD	32	Liver	8
29	[139]	CCS	Wistar rats	PM_2.5_	32	Blood	8
30	[140]	CCS	Wistar rats	PM_2.5_ + HFD	112	Blood and heart	8
31	[141]	CCS	C57BL/6 J mice, Nlrp3	PM_2.5_	52	Macro-phages	8
32	[142]	CCS	Wistar rats	PM_2.5_, HFD, and PM_2.5_ + HFD	32	Liver	8
33	[20]	CCS	C57Bl/6 mice	PM_2.5_ and HFD	20	Adipose tissue	6

Ref. = reference; StD = study design; PBM = population or biological model; n = sample size; TCL = tissue or cell line; QS = quality score; CCS = case–control study; L = longitudinal; T = transversal; C = cohort; NA = Not applicable; HFD = high-fat diet. Studies involving cell lines refer to lipid exposure, and PM_2.5_ = fine particulate matter suspended in the air with a diameter ≤ 2.5 microns.

## Data Availability

The original contributions presented in the study are included in the article and Appendix A. Further inquiries can be directed to the corresponding author.

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
