# Peer review of "Molecular Pathways Linking High-Fat Diet and PM2.5 Exposure to Metabolically Abnormal Obesity: A Systematic Review and Meta-Analysis"

_biomolecules, 2024, doi:10.3390/biom14121607_

Round 1
Reviewer 1 Report
Comments and Suggestions for Authors
Molecular Pathways Linking High-Fat Diet and PM2.5 Exposure to Metabolically Abnormal Obesity: A Systematic Review and 3 Meta-Analysis
The study highlights the alterations caused by exposure to HFD and PM2.5, including increased body weight, oxidative stress, elevated insulin levels, and other metabolic changes. The review presents significant variables affected by metabolically abnormal obesity, but it may be difficult for readers seeking a concise and clear explanation to follow.
Reviewer comments:
-An abbreviation should not be used before it is defined properly (see “PPAR” on lines 40 and 75).
-There is no need for an additional introduction once an abbreviation has been introduced (e.g., Lines 123 and 187).
-Tables and figures referenced within the text should be formatted in bold (e.g., Lines 229, 379, 395, and 577).
-Introducing a new section on the possible future directions of these findings would not only improve the quality of the work but also enhance its impact on the literature and future research. This might be supported with an additional/simplified figure if found necessary.
-The resolution and overall quality of the figures should be improved to ensure a clearer understanding for readers. Additionally, Latin phrases in the figures (e.g., et al.) should be updated to their italicized forms.
-The conclusion part needs to be more extensive. This research article has comprehensive information and it makes that the conclusion part must be more comprehensive such as future outlooks, potential treatments etc.
-Information about PPAR and HIF-1 signaling pathways should be added to the conclusion.
Author Response
Response to Reviewer 1 Comments
We sincerely thank the reviewer for his/her valuable feedback and suggestions. All changes have been incorporated into the manuscript and are highlighted in red for easy reference
Point-by-point response to Comments and Suggestions for Authors
Comments 1: An abbreviation should not be used before it is defined properly (see “PPAR” on lines 40 and 75).
Response 1: Modified. It was added the full definition of “PPAR” on its first appearance. To maintain consistency, we have highlighted this definition in the section where it was already correctly defined, so that its consistency can be verified throughout the text.
Comments 2. There is no need for an additional introduction once an abbreviation has been introduced (e.g., Lines 123 and 187).
Response 2: Done. Regarding abbreviations, all mentions have been reviewed to avoid unnecessary repetitions. We have corrected the introduction of repeated abbreviations and maintained consistency in their use throughout the manuscript.
Comments 3: Tables and figures referenced within the text should be formatted in bold (e.g., Lines 229, 379, 395, and 577).
Response 3: We have changed it. We have reviewed the presentation of the tables and figures cited in the text, highlighting them in bold for greater visibility.
Comments 4: Introducing a new section on the possible future directions of these findings would not only improve the quality of the work but also enhance its impact on the literature and future research. This might be supported with an additional/simplified figure if found necessary.
Response 4: Agree. We omit the perspectives in the conclusion to delve into a new section. We added the section entitle “Prospectives” located between the discussion and the conclusion, as recommended. We considered that an additional figure is not necessary, since it was not considered necessary for the clarity of the section.
Comments 5: The resolution and overall quality of the figures should be improved to ensure a clearer understanding for readers. Additionally, Latin phrases in the figures (e.g., et al.) should be updated to their italicized forms.
Response 5: The images inserted in the original document are of low quality but were delivered in the format required by the magazine, so we are confident that the quality will improve during the editorial process. Regarding the Latin phrases, we have ensured that they are presented in italics, in accordance with the recommendations.
Comments 6: The conclusion part needs to be more extensive. This research article has comprehensive information and it makes that the conclusion part must be more comprehensive such as future outlooks, potential treatments etc.
Response 6: We agree. The conclusion was restructured to make it longer and complete. Information on PPAR and HIF-1 signaling pathways included in the new conclusion.

Reviewer 2 Report
Comments and Suggestions for Authors
General comments
This systematic review and meta-analysis of 33 studies investigates the molecular pathways linking high-fat diet, PM2.5 exposure, and metabolically abnormal obesity. Overall, the article is well-written, and the methods are generally acceptable. The literature is up-to-date. However, several issues are not well described. Table 1 is not comprehensive, the forest plots lack clarity and many abbreviations are not adequately defined or explained, making it challenging to understand the findings and conduct a comprehensive evaluation. Major revision is necessary before further consideration!
As there are numerous abbreviations used in the article, this reviewer strongly suggests including an abbreviation list with their corresponding definitions to enhance clarity and ease of understanding.
Specific comments
Line 37 and 40: Full name of HDL, SOD, PPAR and AMPK should be given.
Line 53-54: The classification of healthy and unhealthy obesity should be introduced. What is the difference between these two terms?
Line 55: Regarding the reference for obesity as risk factor of non-communicable chronic diseases (i.e. [9]) , it might be more appropriate to cite a review article rather than a study focused on a specific country.
Line 57: The term 'environmental pollution' is broad. Therefore, it would be better to specify by adding 'including air pollution' after 'environmental pollution,' as the cited reference [12] pertains only to air pollution.
Line 62-64: The sentence “ Airborne --- climate change” is very long. It might be better to split it into two separate sentences for clarity.
Line 72: Is the term “metabolically abnormal obesity” same as “ metabolically unhealthy obesity” mentioned at line 54?
Line 73: The definition of FAT, CEJUS ,UP and DN should be introduced.
Line 87: For the experimental model, the species should be given.
Line 98: The definition of ABC should be introduced here, instead of at Line 99.
Line 130: The definition of FoxO1 should be introduced.
Line 155- 156: For the statement “ Obesity--- (WAT)”, was this observation found in animal or human? please clearly indicate it in the text.
Line 177, 178, 181, 182, 186, 195 and 197: The abbreviations such as AG, LEP, FTO, GHRL, NRF, HIF, STAT, PHD should be introduced.
Line 280-284: As human studies are included in selected 33 articles, the description of human studies must be described.
Line 280-282: sentence “ The 88% ---- investigation” is not clear. Rephrase is necessary.
Table 1: The exposure and outcomes should also be given in the table as these information are very important for the understanding. In addition, the author should be given at column “ Ref.” because in the subsequent figures, author names are given. Moreover, in 2.4. Quality Assessment it is stated that cohort study and case-control studies were assessed with NOS. For Cross-sectional studies AXIS was used and the quality is given as percentage. However, in Table 1, [119] is a human cohort study and QS is given as 70% while the QS of other human studies (transversal) are given as 8 and 7, separately. Does it mean that QS of two transversal human studies were assessed with NOS? Are transversal human study same as cross-sectional study? Please clarify.
Line 291, 293: Is IC 95% same as 95% CI in Fig. 2? Same question is also for Fig. 3, 4 and 5!
Fig. 2: Please add species in the title of Fig. 2 as only 10-12 mice studies were included in this figure. Additionally, since Table 1 includes 21 studies involving mice, please explain why only these 10-12 mice studies are selected for inclusion in this figure. Please clearly indicate what the estimate represents (coefficient, mean or something else) in the figure and provide their corresponding unit? Moreover, the corresponding reference number should be given after author name in Forest plot. A figure legend to explain A, BB, AB, AD, C, BD given in each study and dash line, etc. must be provided. Same comment also apply for Fig. 3, 4, 5!
Line 316-318: please provide the unit of glucose, insulin and SOD.
Line 338-349: please provide the unit of Total cholesterol, HDL, LDL, and TG.
Line 375: The definition of BAT must be introduced.
Line 412: The definition of NPY, AGRP should be introduced.
Fig. 9: The line colors for Apoa1& Apoa5 and Ghrl in Fig. 9 are similar, making it difficult to distinguish between them.
Fig. 11: More explanation is needed for this figure and the figure should be modified to make it clearer. Additionally, what is the association between upregulated genes or up-& down regulated genes and genes in grey and light blue boxes?
Line 589: Regarding “ Fasn”, in Fig 13, it is given as FAS? Please clarify it.
Author Response
Response to Reviewer 2 Comments
We sincerely thank the reviewer for his/her valuable comments and suggestions. All changes have been incorporated into the manuscript and are highlighted in red for easy reference.
Point-by-point response to Comments and Suggestions for Authors
Comments 1: As there are numerous abbreviations used in the article, this reviewer strongly suggests including an abbreviation list with their corresponding definitions to enhance clarity and ease of understanding.
Response 1: We agree. A list of abbreviations with their corresponding definitions has been added, to improve the clarity and ease of comprehension.
Comments 2: Line 37 and 40: Full name of HDL, SOD, PPAR and AMPK should be given.
Response 2: It was corrected providing the full names of HDL, SOD, PPAR y AMPK.
Comments 3: Line 53-54: The classification of healthy and unhealthy obesity should be introduced. What is the difference between these two terms?
Response 3: The wording was corrected, as it was written whit “o” which generated ambiguity, and the classification of healthy and unhealthy obesity was complemented.
Comments 4: Line 55: Regarding the reference for obesity as risk factor of non-communicable chronic diseases (i.e. [9]) , it might be more appropriate to cite a review article rather than a study focused on a specific country.
Response 4: The change was made, and a review article was cited instead of and study focused on a specific country.
Comments 5: Line 57: The term 'environmental pollution' is broad. Therefore, it would be better to specify by adding 'including air pollution' after 'environmental pollution,' as the cited reference [12] pertains only to air pollution.
Response 5: The modification was made and “including air pollution” was added after “environmental pollution, since the reference cited [12] refers only to pollution.
Comments 6: Line 62-64: The sentence “ Airborne --- climate change” is very long. It might be better to split it into two separate sentences for clarity.
Response 6: We split the sentence “Airborne—climate change “into two separate statements to improve clarity.
Comments 7: Line 72: Is the term “metabolically abnormal obesity” same as “ metabolically unhealthy obesity” mentioned at line 54?
Response 7: By correcting the obesity classification this doubt is clarified.
Comments 8: Line 73: The definition of FAT, CEJUS ,UP and DN should be introduced.
Response 8: The definition of FAT, CEJUS, UP and DN was included to provide clarity and context to the mentioned terms.
Comments 9: Line 87: For the experimental model, the species should be given.
Response 9: The model was added.
Comments 10: Line 98: The definition of ABC should be introduced here, instead of at Line 99.
Response 10: The definition was added.
Comments 11: Line 130: The definition of FoxO1 should be introduced.
Response 11: The definition was added.
Comments 12: Line 155- 156: For the statement “ Obesity--- (WAT)”, was this observation found in animal or human? please clearly indicate it in the text.
Response 12: It was clarified that the observation in obesity and white adipose tissue (WAT) corresponds to studies in murine models.
Comments 13: Line 177, 178, 181, 182, 186, 195 and 197: The abbreviations such as AG, LEP, FTO, GHRL, NRF, HIF, STAT, PHD should be introduced.
Response 13: Abbreviations were introduced and defined.
Comments 14: Line 280-284: As human studies are included in selected 33 articles, the description of human studies must be described.
Response 14: The description of these studies was added.
Comments 15: Line 280-282: sentence “ The 88% ---- investigation” is not clear. Rephrase is necessary.
Response 15: The redaction was changed to improve clarity.
Comments 16: Table 1: The exposure and outcomes should also be given in the table as these information are very important for the understanding.
Response 16: The exposures were added to the table, but no the results, since we searched for several variables and the table would become saturated with information. However, the contents from which the information was extracted are available in the Open Science Framework, as recorded when conducting the systematic review and meta-analysis as requested by the platform.
Comments 17: In addition, the author should be given at column “ Ref.” because in the subsequent figures, author names are given.
Response 17: Yes, we are in table 1, we are item 33.
Comments 18: Moreover, in 2.4. Quality Assessment it is stated that cohort study and case-control studies were assessed with NOS. For Cross-sectional studies AXIS was used and the quality is given as percentage. However, in Table 1, [119] is a human cohort study and QS is given as 70% while the QS of other human studies (transversal) are given as 8 and 7, separately. Does it mean that QS of two transversal human studies were assessed with NOS? Are transversal human study same as cross-sectional study? Please clarify.
Response 18: We have reviewed the table with the evaluations of each article and noticed a modification in the Quality Score (QS) of four articles, which occurred while the file was being edited collaboratively in a shared drive. The corrections have now been made.
Comments 19: Line 291, 293: Is IC 95% same as 95% CI in Fig. 2? Same question is also for Fig. 3, 4 and 5!
Response 19: It is CI, we already have corrected all the errors.
Comments 20: Fig. 2: Please add species in the title of Fig. 2 as only 10-12 mice studies were included in this figure. Additionally, since Table 1 includes 21 studies involving mice, please explain why only these 10-12 mice studies are selected for inclusion in this figure. Please clearly indicate what the estimate represents (coefficient, mean or something else) in the figure and provide their corresponding unit?
Response 20: We did not use all articles for analysis because not all studies reported all the variables included in the present study. Description of symbology was added, which includes the measurements.
Comments 21: A figure legend to explain A, BB, AB, AD, C, BD given in each study and dash line, etc. must be provided.
Response 21: The program used did not allow the use of a numerical system for references, nor the same nomenclature of the study for different exhibitions because it marketed it as a repeated value, so letters were added, in this sense, the numerical reference column was manually added to the figures. Also, it was re-attached to the files as a figure.
Comments 22: Same comment also apply for Fig. 3, 4, 5!
Response 22: Pertinent corrections were made.
Comments 23: Line 316-318: please provide the unit of glucose, insulin and SOD.
Response 23: Units were added.
Comments 24: Line 338-349: please provide the unit of Total cholesterol, HDL, LDL, and TG.
Response 24: Units were added.
Comments 25: Line 375: The definition of BAT must be introduced.
Response 25: The definition was added.
Comments 26: Line 412: The definition of NPY, AGRP should be introduced.
Response 26: The definition was added.
Comments 27: Fig. 9: The line colors for Apoa1& Apoa5 and Ghrl in Fig. 9 are similar, making it difficult to distinguish between them.
Response 27: The colors have been changed, now they are violet and blue.
Comments 28: Fig. 11: More explanation is needed for this figure and the figure should be modified to make it clearer. Additionally, what is the association between upregulated genes or up-& down regulated genes and genes in grey and light blue boxes?
Response 28: We added the explanation in the figure caption.
Comments 29: Line 589: Regarding “ Fasn”, in Fig 13, it is given as FAS? Please clarify it.
Response 29: We added the clarification in the figure caption.

Round 2
Reviewer 1 Report
Comments and Suggestions for Authors
I thank the authors for their responses. This version of the article is suitable for publication.